# Mother cells control daughter cell proliferation in intestinal organoids to minimize proliferation fluctuations

**Guizela Huelsz-Prince[1], Rutger Nico Ulbe Kok[1], Yvonne Goos[1], Lotte Bruens[2], Xuan Zheng[1], Saskia Ellenbroek[2], Jacco Van Rheenen[2], Sander Tans[1], Jeroen S van Zon[1]***

[1]AMOLF, Science Park, Amsterdam, Netherlands; [2]Department of Molecular Pathology, Oncode Institute, Netherlands Cancer Institute, Amterdam, Netherlands

**Abstract** During renewal of the intestine, cells are continuously generated by proliferation. Proliferation and differentiation must be tightly balanced, as any bias toward proliferation results in uncontrolled exponential growth. Yet, the inherently stochastic nature of cells raises the question how such fluctuations are limited. We used time-lapse microscopy to track all cells in crypts of growing mouse intestinal organoids for multiple generations, allowing full reconstruction of the underlying lineage dynamics in space and time. Proliferative behavior was highly symmetric between sister cells, with both sisters either jointly ceasing or continuing proliferation. Simulations revealed that such symmetric proliferative behavior minimizes cell number fluctuations, explaining our observation that proliferating cell number remained constant even as crypts increased in size considerably. Proliferative symmetry did not reflect positional symmetry but rather lineage control through the mother cell. Our results indicate a concrete mechanism to balance proliferation and differentiation with minimal fluctuations that may be broadly relevant for other tissues.

*For correspondence:
j.v.zon@amolf.nl

**Competing interest:** The authors declare that no competing interests exist.

## Editor's evaluation

This paper is a fundamental work in developmental biology that supports its findings with compelling evidence drawn from both theoretical and experiment insights. It provides a potentially general mechanism for the control of a proliferative cell population. This work will be of interest to researchers in the fields of developmental and stem cell biology.

## Introduction

Most adult organs and tissues constantly renew themselves by replacing old and damaged cells, while retaining their structure (*Simons and Clevers, 2011*). Theory indicates that this homeostasis requires a precise balance between proliferating and non-proliferating cells, as even a slight systematic bias toward producing proliferating cells yields uncontrolled exponential cell growth (*Lander et al., 2009*; *Clayton et al., 2007*; *Klein et al., 2007*; *Klein and Simons, 2011*; *Lopez-Garcia et al., 2010*; *Rué and Martinez Arias, 2015*). Moreover, the exponential nature of proliferation also readily amplifies fluctuations in the number of proliferating cells, which can lead to stochastic cell overgrowth or depletion in the absence of additional control mechanisms (*Feller, 1939*; *Sun and Komarova, 2012*). How cell proliferation is balanced despite fluctuations has remained challenging to test in direct experiments, given the difficulties of following this process in time.

The mammalian intestine has become an important model system to study the mechanisms of tissue renewal and homeostasis (*Simons and Clevers, 2011*; *Gehart and Clevers, 2019*). The proliferating

**eLife digest** The vast majority of cells lining our intestine die within three to five days. They are replaced by a small group of stem cells which divide to produce either more stem cells, or cells that stop dividing and transform, or 'differentiate', in to mature cells in the intestine. Stem cells must generate the same number of dividing and differentiated cells. If there is even a slight bias and too many stem cells are produced, this can lead to uncontrolled growth, which is the root cause of cancer.

In principal, the best way to achieve this balance is for stem cells to always asymmetrically divide in to two distinct cells: one that will continue to divide, and another that will mature in to an adult cell. However, recent research suggests that this process is much more random, with stem cells also dividing symmetrically, either in to two stem cells or two differentiated cells. So, how does the random nature of stem cell divisions not cause the number of dividing cells to fluctuate unpredictably in the intestine?

To investigate, Huelsz-Prince et al. studied stem cells in a miniature model of the mouse intestine, known as an organoid, which can be grown outside of the body in a laboratory. All stem cells and their progeny were tracked for over 65 hours using a microscope to see how many dividing and differentiated cells they formed. This revealed that almost all stem cells in the organoid split symmetrically rather than asymmetrically.

Huelsz-Prince et al. then developed a computer model of stem cells in the model intestine and tested the impact of changing the proportion of symmetric and asymmetric divisions. The results showed that having more symmetric divisions reduced fluctuations in the number of dividing cells better than high levels of asymmetric divisions.

Other organs rely on a similar system to the intestine to replenish their mature cells. Consequently, the finding that symmetric divisions control fluctuations in the number of stem cells may be applicable to other parts of the body. Further testing with human disease samples, such as cells from cancer patients, using the organoid model system may also shed light on how division is disrupted in these conditions.

stem cells that sit at the base of intestinal crypts generate rapidly dividing transit-amplifying (TA) cells that in turn replenish the absorptive and secretory cells populating the lining of intestinal villi. Paneth cells positioned at the crypt bottom provide short-range signals that affect the proliferative and undifferentiated state of intestinal stem cells (*Farin et al., 2012*; *Sato et al., 2011*; *Shoshkes-Carmel et al., 2018*). Originally, it was proposed that one or a few stem cells generated all differentiated cells by strictly asymmetric cell divisions (*Scoville et al., 2008*; *Winton and Ponder, 1990*), thus directly ensuring a constant stem cell pool. Subsequent studies rather suggested that individual cells stochastically and independently cease to divide or not (*Lopez-Garcia et al., 2010*; *Snippert et al., 2010*; *Ritsma et al., 2014*). In this 'population asymmetry' model, in principle, one stem cell daughter could remain proliferative by staying adjacent to a Paneth cell, while the other daughter exits the stem cell niche, differentiates, and stops proliferating. However, such asymmetric outcome is no longer guaranteed. Instead, proliferation and differentiation are balanced more indirectly, by averaging these stochastic events across the total stem cell population.

Observations of neutral drift, in which the offspring of a single cell randomly takes over the stem cell population of intestinal crypts (*Lopez-Garcia et al., 2010*; *Snippert et al., 2010*; *Ritsma et al., 2014*) established the stochastic nature of stem cell proliferation that distinguishes the population asymmetry model from the earlier division asymmetry model. However, approaches used thus far do not follow the underlying cell divisions and lineages in time. Proliferation symmetry between sister cells and its role in homeostasis of the intestinal epithelium has so far only been inferred indirectly, typically by quantifying the clone size distributions at a certain time point. Hence, we also lack insight into the fluctuations in the number of proliferating cells and the mechanisms that control them.

Here, we developed an alternative approach: we employed time-lapse microscopy and single-cell tracking of all cells in crypts of mouse intestinal organoids (*Sato et al., 2009*), thus providing complete lineage trees, division dynamics, and cell movement, and combine it with mathematical modeling and intravital imaging of the mouse intestine. Surprisingly, we found that most cell divisions (>90%) were symmetric in proliferative outcome, producing daughter cells that either both continued to proliferate

or both ceased proliferating. Proliferation was symmetric even when one daughter neighbored a Paneth cell, the source of proliferative signals in the crypt, while the other did not. Hence, proliferation was not independent between sisters but rather controlled through the lineage by the mother. Our data and simulations explained not only how this behavior achieves homeostasis, but moreover, that it constitutes a near-optimal strategy to minimize fluctuations in the number of proliferating cells. Consistently, despite their large size increases over multiple generations in crypts of various sizes, the number of proliferating cells was notably constant in time and exhibited sub-Poissonian fluctuations, indicating a precise balance between proliferative and non-proliferative sister pairs. Finally, by measuring clone size distributions in mice, we showed that stem cell divisions in vivo reproduced the strong symmetry in proliferative behavior between sister cells seen in organoids. As cell proliferation in many tissues follows inherently stochastic 'population asymmetry' mechanisms (*Clayton et al., 2007*; *Snippert et al., 2010*; *Doupé et al., 2012*; *Klein et al., 2010*; *Teixeira et al., 2013*), we conjecture that high symmetry in proliferative behavior, controlled through the lineage, may be a more general mechanism to limit proliferation fluctuations.

## Results

### Single-cell tracking of complete crypts in growing intestinal organoids

To examine the dynamics of individual cells within crypts during growth, we used organoids with a H2B-mCherry nuclear reporter (*Figure 1A*) and performed confocal three-dimensional (3D) time-lapse microscopy for up to 65 hours at a time resolution of 12 minutes. Cell division events were distinguished by the apical displacement of the mother cell nucleus, followed by chromosome condensation and separation, and finally, basal migration of the daughter cell nuclei (*Figure 1B*), consistent with epithelial divisions (*Ragkousi and Gibson, 2014*). Custom-written software (*Kok and van Zon, 2016*) was used to track every cell within organoid crypts by recording their nuclei positions in 3D space and time (*Figure 1C*, *Figure 1—video 1*) and reconstruct lineage trees containing up to six generations (*Figure 1F*, *Figure 1—figure supplement 1*).

To study the cell lineages along the crypt surface, we 'unwrapped' the crypts: first, we annotated the crypt axes at every time point, then projected every cell position onto the surface of a corresponding cylinder, which we then unfolded (*Figure 1D and E*). This allowed us to visualize the cellular dynamics in a two-dimensional plane defined by two coordinates: the position along the axis and the angle around the axis. We found that lineages starting close to the crypt bottom typically continued to proliferate and expand in cell number, while those further up in the crypt contained cells that no longer divided during the experiment, consistent with stem cells being located at the crypt bottom and terminal differentiation occurring higher up along the crypt axis (*Figure 1F*). Lineages were also terminated by the death of cells, as observed by their extrusion into the lumen. This fate occurred more often in some lineages compared to others, and the frequency did not depend strongly on position along the crypt-villus axis. At the crypt bottom we also observed a small number of non-dividing cells, suggestive of terminally differentiated Paneth cells. Indeed, these cells typically exhibited the larger cell size and granules typical of Paneth cells. Finally, a small fraction of cells could not be tracked during the experiment, as they moved outside of the field of view, or their fluorescence signal was degraded due to scattering in the tissue.

### Control of cell proliferation in organoid crypts

To systematically study proliferation control, we classified cells as proliferating when they divided during the experiment. Cells were classified as non-proliferating when they did not divide for >30 hr or were born >60 μm away from the crypt bottom, as such cells rarely divided in our experiments (see Materials and methods for details). A smaller fraction of cells could not be classified. Some cells were lost from tracking (7%, N=2880 cells). These cells were typically located in the villus region (*Figure 1F*) and therefore likely non-proliferating. For other cells, the experiment ended before a division could be observed or excluded based on the criteria above (27%). Such cases were particularly prominent in the last 15 hr of each time-lapse data set (22%). To analyze proliferation control, we therefore excluded all cells born <15 hr before the end of each experiment, thereby reducing the fraction of unclassified cells to 10%.

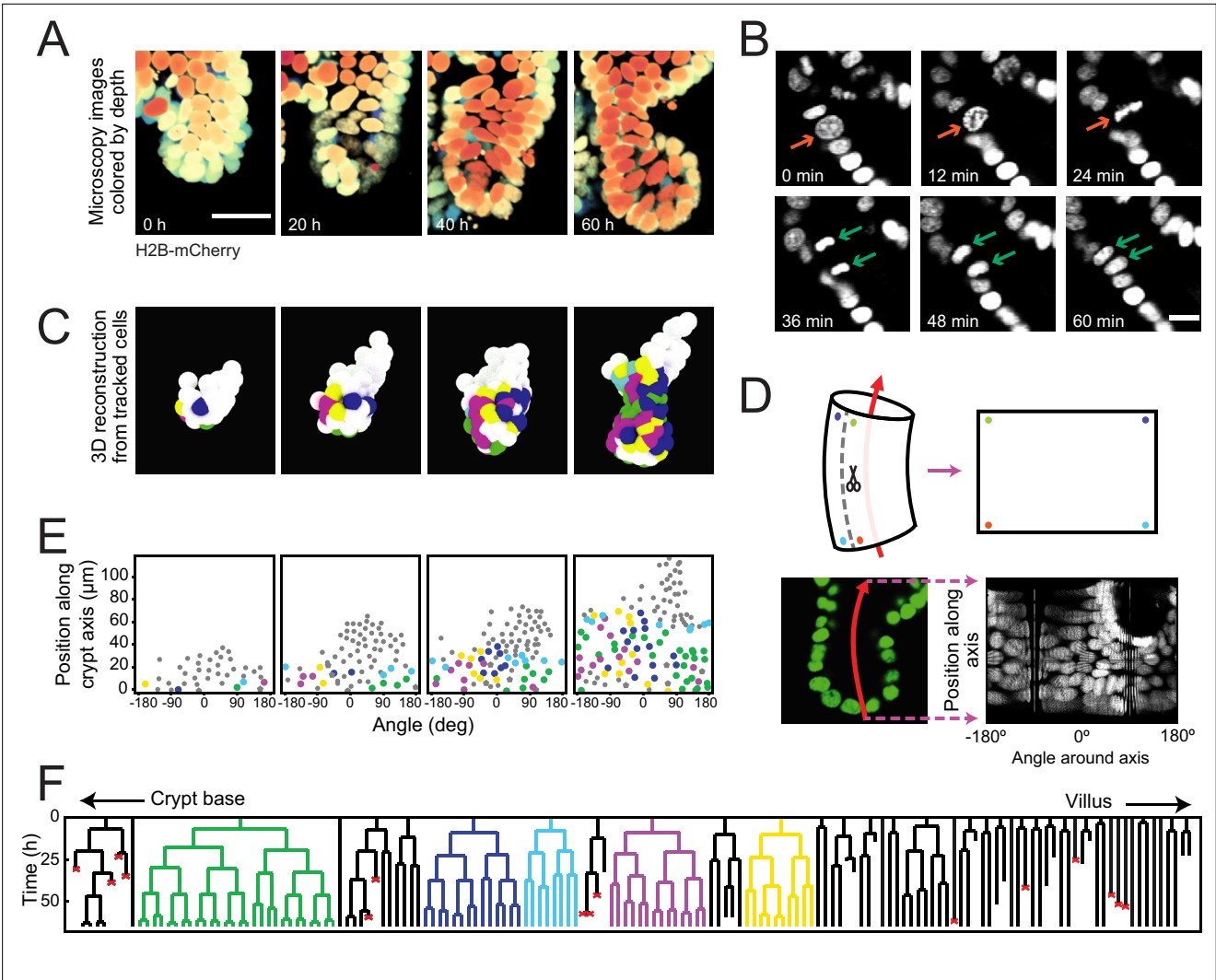

**Figure 1.** Time-lapse imaging and single-cell tracking of intestinal organoid crypts. (**A**) Three-dimensional (3D) reconstruction of an organoid expressing an H2B-mCherry reporter to visualize individual nuclei. Shown here is the crypt region, with nuclei colored by their depth along the optical axis. Scale bar is 25 µm. (**B**) Snapshots of a cell division event in a crypt. Cell divisions are distinguished by the apical migration of the nucleus followed by chromosome condensation (red arrows). After mitosis, the nuclei of the two newly born cells are displaced basally (green arrows). Scale bar is 10 µm. (**C**) 3D reconstruction of a crypt growing in time using the positions of tracked nuclei. Colors represent cells that belong to the same lineage. (**D**) Illustration of crypt unwrapping. After the crypt-villus axis is annotated (red arrow), tracked cell positions are projected onto the surface of a bent cylinder. The cylinder is then unfolded, and its surface is mapped onto a two-dimensional plane defined by the distance along the axis and the angle around the axis. (**E**) Unwrapped representation of the crypt in (**C**), where colors represent the same lineages. (**F**) Lineage trees of cells within the crypt in (**C**) and colored accordingly. Cells in the initial time point are ordered according to their distance to the crypt base. Red crosses indicate cell deaths, and incomplete lines indicate cells that could not be accurately traced further due to insufficient fluorescence intensity or movement outside of the field of view.

The online version of this article includes the following video and figure supplement(s) for figure 1:

**Figure supplement 1.** Lineage trees of all tracked cells.

**Figure 1—video 1.** Tracking cell position and lineage.

https://elifesciences.org/articles/80682/figures#fig1video1

Using this classification procedure, we then quantified the total number of cells born in the tracked cell lineages for nine crypts and found a strong (~fourfold) increase in time (*Figure 2A*). In contrast, the number of proliferating cells remained approximately constant in time for most crypts (*Figure 2B*). Two crypts (crypts 3 and 4 in *Figure 2*) formed an exception with ~twofold increase in the number of proliferating cells, an observation that we discuss further below. We then estimated the exponential growth rate $\alpha$ for each crypt, by fitting the dynamics of total number of cells born and proliferating

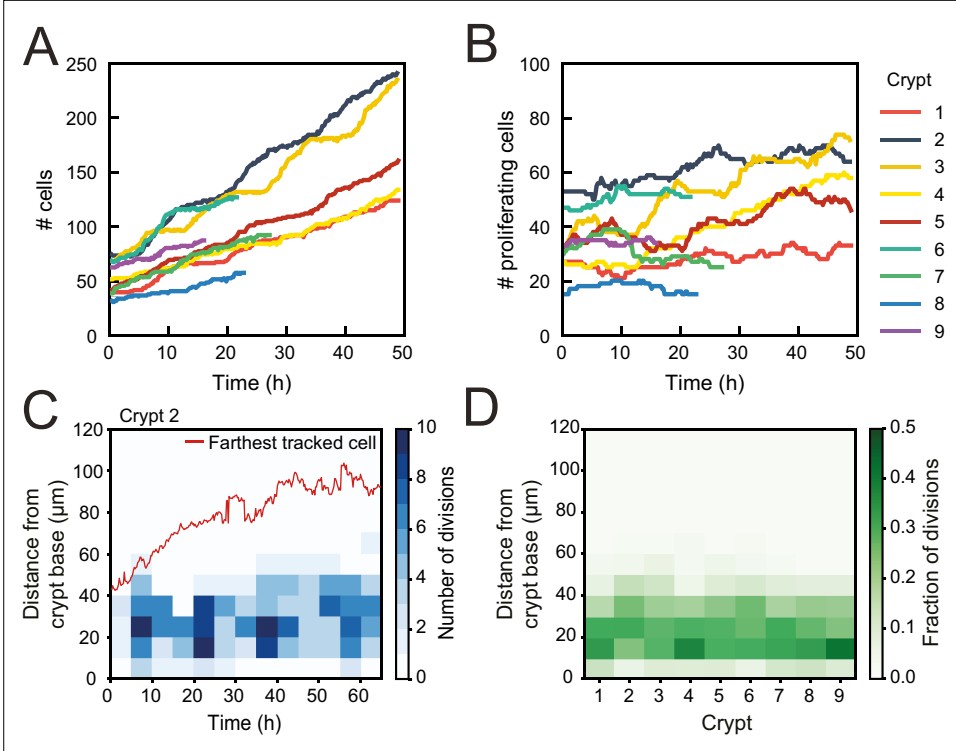

**Figure 2.** Control of cell divisions in intestinal organoids. (**A**) Total number of cells born and (**B**) number of proliferating cells as a function of time for all cell lineages followed in nine tracked crypts. Cells that died were classified as non-proliferating. Note the different scales along the y-axis. Whereas total cell number increases, the number of proliferating cells remains approximately constant. The strongest increase in number of proliferating cells (~twofold) was seen in crypts 3 and 4. (**C**) Number of divisions that occurred at different positions along the crypt axis as a function of time in a single-tracked crypt. Red line corresponds to the position of the farthest tracked cell from the crypt base at every time point. Divisions occur in a compartment close to the crypt base, whose size remains constant over time. Apical displacement of the nuclei during mitosis results in few divisions occurring at less than 10 μm from the crypt base. (**D**) Fraction of divisions that occurred at different positions along the crypt axis for all tracked crypts, averaged over the full-time course. The size of the proliferative region is similar between crypts, despite differences in the total number of divisions.

The online version of this article includes the following figure supplement(s) for figure 2:

**Figure supplement 1.** Crypt growth and heterogeneity.

cell number to a simple model of cell proliferation (discussed further below as the Uniform model), where proliferating cells divide randomly into proliferating and non-proliferating cells. In this model, the number of proliferating and non-proliferating cells increases on average by $\alpha$ and $1-\alpha$ per cell division, respectively (Materials and methods). Apart from crypts 3 and 4, that displayed growth ($\alpha$), the remaining crypts showed a low growth rate, $\alpha=0.05$ (*Figure 2—figure supplement 1A–C*), indicating that birth of proliferating and non-proliferating cells was balanced on average. We then quantified the magnitude of fluctuations in the number of proliferating cells, $N$. Calculations of birth-death models of cell proliferation show that, without any control, the standard deviation of the proliferating cell number grows in time without bounds as $\sigma_D\sqrt{Nt}$ (*Feller, 1939*). In models without exponential growth, with proliferating cells born at constant rate, fluctuations are reduced: they are constant in time and Poissonian, $\sigma_D\sqrt{N}$ (*Swain, 2016*). In models where exponential growth was controlled by homeostatic feedback loops, fluctuations were further reduced to sub-Poissonian: $\sigma_D < \sqrt{N}$ (*Sun and Komarova, 2012*). Using the same measures, we found here that most crypts exhibited sub-Poissonian fluctuations (*Figure 2—figure supplement 1D*), implying the presence of a mechanism to limit fluctuations in proliferating cell number. Finally, we quantified the frequency of cell divisions along the crypt axis. Notably, divisions occurred in a region below 60 μm from the crypt base throughout the experiment, even as the crypts grew significantly (*Figure 2C*), indicating that the size of the proliferative region

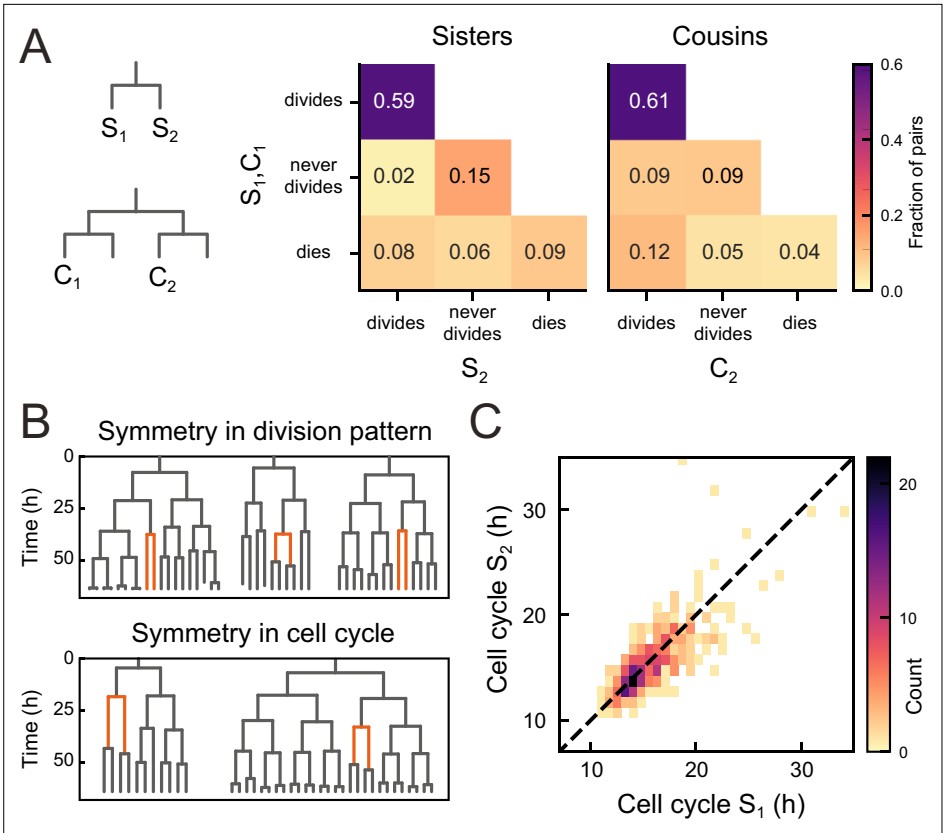

**Figure 3.** Symmetry of lineage dynamics between sister cells. (**A**) Correlations in division patterns between sister ($S_1$,$S_2$) and cousin cells ($C_1$,$C_2$) (n=1004 and 1304 sister and cousin cell pairs). Most sister pairs show symmetrical outcomes, with most pairs consisting of sisters that either both divide or both cease dividing. Cell death occurs at low frequency and impacts sister cells asymmetrically. Symmetrical outcomes are still dominant for cousins, but the fraction of pairs that exhibited asymmetric proliferative outcomes ($C_1$ never divides, $C_2$ divides) was significantly increased in cousins compared to sisters (p=2.4·10$^{-7}$, Pearson's Chi Square test). (**B**) Representative examples of measured lineages highlighting pairs of sister cells (orange) that differ in lineage dynamics from their more distant relatives (black), either in terms of proliferative behavior (top) or cell cycle duration (bottom). (**C**) Duration of sister cell cycles plotted against each other for pairs in which both sisters divided. Cell cycle duration is strongly correlated between sisters (R=0.80).

The online version of this article includes the following figure supplement(s) for figure 3:

**Figure supplement 1.** Cell cycle distribution.

**Figure supplement 2.** Sister pair division patterns and proliferation control.

was constant in time. Moreover, the proliferative region was found to have a similar size in all analyzed crypts (*Figure 2D*), even though crypts varied both in size, as measured by diameter (30–50 µm, *Figure 2—figure supplement 1E*), and number of proliferating cells (*Figure 2B*). Overall, these results show that crypts by themselves are already capable of a specific form of homeostasis, namely, maintaining a stationary number of proliferating cells that occupy a region of the crypt of constant size.

## Symmetry of proliferative behavior between sister cells

To examine the origin of the observed balance between the birth of proliferating and non-proliferating cells, we first examined whether cell proliferation or cell death was correlated between sisters, for all observed sister pairs $S_1$ and $S_2$ (*Figure 3A*). Strikingly, we found that the decision to divide or not was highly symmetrical between sisters. In particular, occurrences where one sister divided but the other not were rare (2%) compared to cases where both divided or stopped dividing (74%). This correlation was also apparent by visual inspection of individual lineages, as sisters showed the same division behavior (*Figure 3B*, top). Indeed, if we ignore cell death, the fraction of pairs with symmetric

proliferative outcome was high (97%) and could not be explained by sister cells making an independent decision to proliferate or not [(p<10⁻⁵, bootstrap simulation, Materials and methods]. We also compared lineage dynamics between all cousin pairs $C_1$ and $C_2$ (*Figure 3A*). While we indeed found a significantly increased fraction (9%) of cousin pairs with asymmetrical division outcome, i.e., $C_2$ dividing and $C_1$ not, compared to sister pairs (2%), this fraction was still low, indicating that symmetric outcomes also dominated for cousins.

We found that symmetry between sisters did not only impact proliferation arrest, but also cell cycle duration: when a cell exhibited a longer-than-average cell cycle, this was typically mirrored by a similar lengthening of the cell cycle of its sister (*Figure 3B*, bottom). Indeed, cell cycle duration was strongly correlated between sisters ($R$=0.8, *Figure 3C*), even as cell cycle duration showed a broad distribution among tracked cells (*Figure 3—figure supplement 1*). In contrast, cell death was not symmetric between sisters, as the fraction of pairs where both cells died (9%) was smaller than the fraction of pairs where only a single sister died (14%, *Figure 3A*).

When examining all sister pairs in our data set, pairs of dividing sisters (59%) outnumber pairs of non-dividing sisters (15%), which appeared at odds with the observation that in most crypts the number of proliferating cells remains approximately constant (*Figure 2B*). This apparent mismatch was due to the exclusion of sister pairs where the proliferative state could not be classified in one sister or both (*Figure 3—figure supplement 2*), as the majority of these unclassified cells were likely non-proliferating. Indeed, when we restricted our sister pair analysis to the cells of crypts with α≈0 in *Figure 2A and B* (crypts 1, 2, 5, 7–9), and furthermore, assumed that all unclassified cells were non-proliferating, we found that now proliferating sister pairs (43%) are approximately balanced by non-proliferating sisters (40%, *Figure 3—figure supplement 2*).

## Symmetry between sisters minimizes fluctuations in a cell proliferation model

In principle, any combination of (a)symmetric divisions would yield a constant number of proliferating cells on average, as long as the birth of proliferating and non-proliferating cells is balanced. We therefore hypothesized that the observed dominance of symmetric divisions might have a function specifically in controlling *fluctuations* in the number of proliferating cells. To test this hypothesis, we used mathematical modeling. Mathematical models of intestinal cell proliferation have been used to explain observed clone size statistics of stem cells (*Lopez-Garcia et al., 2010*; *Snippert et al., 2010*; *Ritsma et al., 2014*; *Corominas-Murtra et al., 2020*) but so far not to examine the impact of division (a)symmetry on cell number fluctuations. We therefore examined simple stem cell models in which the degree of symmetry of sister cell proliferation could be tuned as an external parameter.

We first examined this in context of the canonical stochastic stem cell model (*Clayton et al., 2007*), that we here refer to as the Uniform model. This model only considers cells as 'proliferating' or 'non-proliferating' (approximating stem and differentiated cells), and tissues that are unbounded in size, while ignoring spatial cell distributions. The parameter $\phi$ describes the division symmetry, with $\phi = 1$ corresponding to purely symmetric divisions and $\phi = 0$ to purely asymmetric divisions. The growth rate $\alpha$ describes the proliferation bias, with $\alpha > 0$ indicating more proliferating daughters, and $\alpha < 0$ more non-proliferating daughters on average. In our simulations, cells either divide symmetrically to produce two proliferating cells with probability $\frac{1}{2}(\phi + \alpha)$ or two non-proliferating cells with probability $\frac{1}{2}(\phi - \alpha)$, while the probability to divide asymmetrically is $1 - \phi$ (*Figure 4A*). The number of proliferating cells increases exponentially for $\alpha > 0$ or decreases exponentially for $\alpha < 0$ while homeostasis requires $\alpha = 0$ (*Lander et al., 2009*; *Clayton et al., 2007*; *Figure 4B*).

When varying the division symmetry Φ while maintaining α=0, we found that fluctuations in the number of proliferating cells $N$ were minimized for Φ=0 (*Figure 4B and C*), i.e., every division was asymmetric. In this scenario, the number of proliferating cells remains constant throughout each individual division by definition. Adding only a small fraction of symmetric divisions strongly increased the fluctuations in $N$. These fluctuations increased the risk of stochastic depletion, where all proliferating cells are lost, or uncontrolled increase in cell number, as previously observed in simulations (*Sun and Komarova, 2012*), with the probability of such events occurring increasing with Φ (*Figure 4B and C*). These trends are inconsistent with the symmetry between sisters we observed experimentally.

Hence, we extended the model by explicitly incorporating the observed subdivision of the crypt in a stem cell niche region, corresponding roughly to the stem cell niche, and a differentiation region,

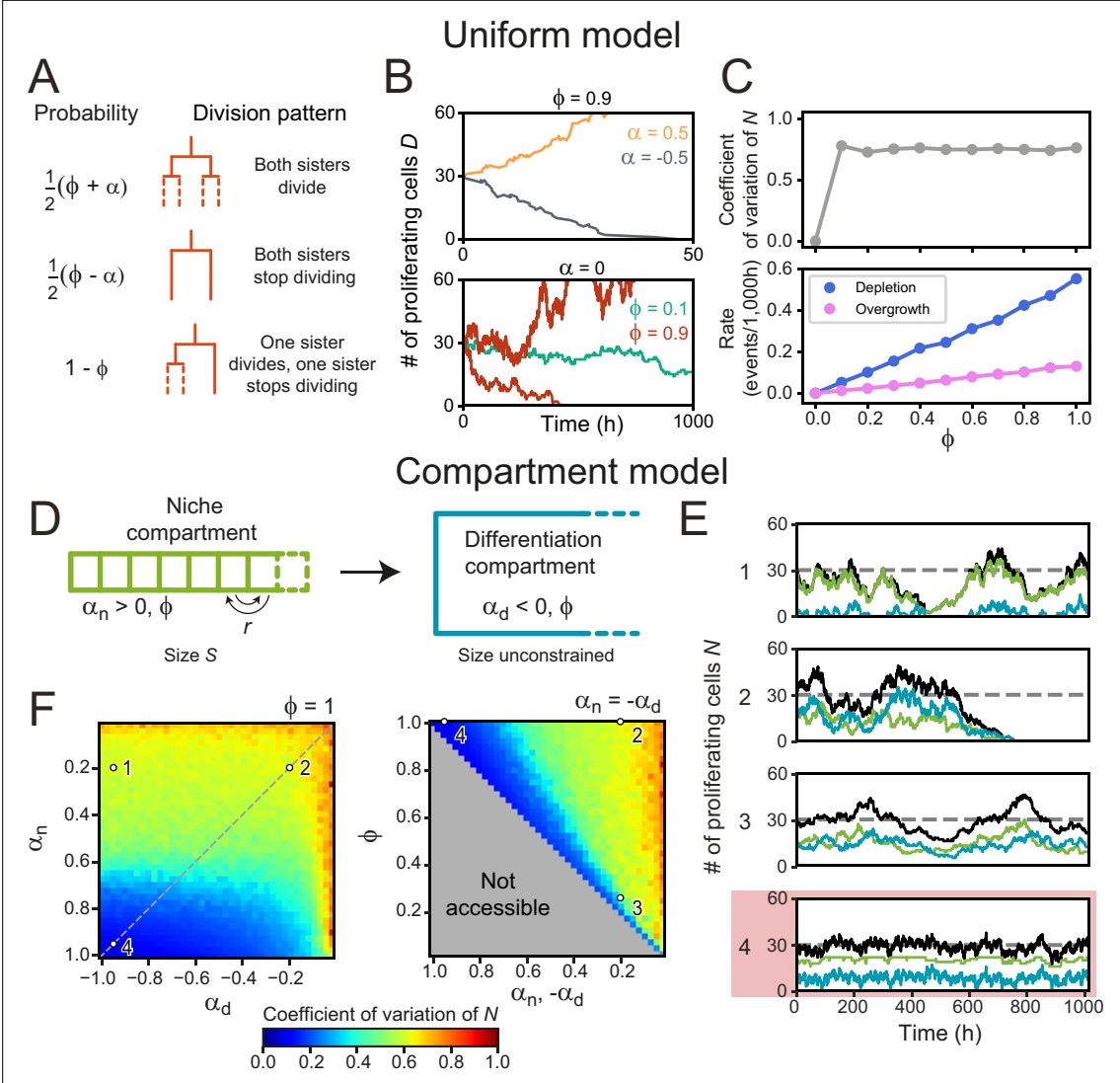

**Figure 4.** Cell number fluctuations in stem cell models. (**A**) 'Uniform' stem cell model. The probability of each division pattern depends on $\alpha$, the average increase in the number of proliferating cells per division, and, $\phi$, the fraction of divisions with symmetric outcome, while the total cell number is unconstrained. (**B**) Number of proliferating cells, $N$ as function of time for different values of $\alpha$ (top) and $\phi$ (bottom). For, $a = 0$, $N$ remains constant on average, yet in this case, fluctuations can cause stochastic depletion or overgrowth of proliferating cells, as shown for $\phi = 0.9$ (bottom). (**C**) Coefficient of variation (standard deviation divided by mean) of $N$ as a function of $\phi$ (top panel) and the probability of depletion ($N = 0$, blue) or overgrowth ($N$ 150, pink) events (bottom panel), for the 'Uniform' model with $a = 0$. Frequency of overgrowth depends strongly on the threshold value used. Fluctuations are minimal for $\phi = 0$, i.e., only asymmetric divisions. (**D**) 'Compartment' model. Cells divide according to (**A**), but now the tissue is divided in a niche compartment $n$, with $\alpha_n > 0$, and a differentiation compartment $d$, where $\alpha_d < 0$. Both compartments have the same $\phi$. In the niche compartment, the total number of cells cannot exceed $S$, so that upon cell division the distalmost cell (dashed square) moves into the differentiation compartment. Cells in the niche compartment switch positions at rate $r$. (**E**) Number of proliferating cells as a function of time in the niche (green) and differentiation compartment (blue). The total number of proliferating cells (black) fluctuates around the dashed line corresponding to $\langle N \rangle$ . Each panel's number refers to the parameter sets shown in (**F**). The parameter set with lowest fluctuations is outlined in red. (**F**) Coefficient of variation of $N$. Left panel shows the effects of varying the growth rates of both compartments when all divisions are symmetric ($\phi = 1$), and right panel of varying the degree of symmetry when both compartments have opposite growth rates ( $\alpha_n = -\alpha_d$ , dashed line in top panel). The gray region in bottom panel is inaccessible parameter space. Simulations ran with $\langle N \rangle = 30$, corresponding to our experimental observations, and rearrangements occurring approximately once per cell cycle. Fluctuations are minimized for $\alpha_n, -\alpha_d = 1$ and $\phi = 1$, i.e., only symmetric divisions.

The online version of this article includes the following figure supplement(s) for figure 4:

**Figure supplement 1.** Two-compartment model.

**Figure supplement 2.** Dependence of lineage dynamics on cell rearrangements.

corresponding to the villus domain (*Figure 4D*). In the niche compartment, which has fixed size $S$, most divisions generate two proliferating daughter cells ($\alpha_n > 0$), while in the differentiation compartment, which has no size constraints, most divisions yield two non-proliferative daughters ($\alpha_d < 0$). Cell divisions in the niche compartment result in expulsion of the distalmost cell into the differentiation compartment, while neighboring cells swap positions in the niche compartment with rate $r$ to include cell rearrangements. In contrast to the uniform model, where homeostasis only occurred for $\alpha = 0$, the compartment model shows homeostasis with $\alpha_{n,d}$ in either compartment. Specifically, we found that the average number of proliferating cells in the two compartments, $N_n$ and $N_d$, is given by $D_n = \alpha_n S$ and $N_d = Sln\left(1 + \alpha_n\right)\frac{\alpha_d \alpha_n}{\alpha_d}\alpha_n S$, independent of the division symmetry $\Phi$ (*Kok et al., 2022*). We simulated the proliferation dynamics for different values of $\alpha_n$, $\alpha_d$, and $\Phi$ (*Figure 4E and F*), where for simplicity we assumed the same $\Phi$ in both compartments. For each combination of parameters, we varied the compartment size $S$ so that $\langle N \rangle = N_n + N_d = 30$, comparable to the number of proliferating cells per crypt in our experiments (Materials and methods, *Figure 4—figure supplement 1A*). For cell rearrangements, we used $r$, where $T$ is the average cell cycle time, meaning that cells rearrange approximately once per cell cycle. For this $r$, our simulations reproduced the correlations in division outcome that we observed experimentally for cousins (*Figure 4—figure supplement 2A–C*), although we found that the dependence of the dynamics of $N$ on the parameters $\phi, \alpha_n$, and $\alpha_d$ did not depend strongly on $r$ (*Figure 4—figure supplement 2D*).

By fixing $\langle N \rangle$, all simulations maintained the same number of proliferating cells on average but potentially differed in the magnitude of fluctuations. Indeed, we found parameter combinations that generated strong fluctuations (*Figure 4E and F*, scenario 1) and stochastic depletion of all proliferating cells (scenario 2). Stochastic depletion occurred at significant rate (>1 event per $10^3$ hr) when $\alpha_n \lesssim 0.5$ (*Figure 4—figure supplement 1B*) and implies that the existence of a stem cell niche, defined as a compartment with $\alpha > 0$, by itself does not guarantee homeostasis, unless its bias toward proliferation is high, $\alpha \approx 1$. For fixed $\alpha_d$ and $\alpha_n$, we observed that fluctuations in $N$ always decreased with more asymmetric divisions ($\Phi \rightarrow 0$) (*Figure 4E and F*, scenarios 2,3), similar to the uniform model. However, the global fluctuation minimum was strikingly different (*Figure 4E and F*, scenario 4). Here, symmetric divisions dominated ($\Phi = 1$), with all divisions generating two proliferative daughters in the niche compartment ($\alpha_n = 1$) and two non-proliferating daughters in the differentiation compartment ($\alpha_d = -1$). Similar low fluctuations were found for a broader range of $\alpha_d$, provided that $\alpha_n \approx 1$. When we quantified the magnitude of fluctuations versus the average number of proliferating cells, we found that fluctuations for the suboptimal scenarios 1–3 were larger than those expected for a Poisson birth-death process (*Figure 2—figure supplement 1D*). In contrast, fluctuations in the optimal scenario 4 were similar to the low fluctuations we observed experimentally.

Our simulations also provided an intuitive explanation for this global minimum. A bias $\alpha_n = 1$ is only reached when the birth of non-proliferating cells in the niche compartment, by symmetric or asymmetric divisions, is fully avoided. In this limit, all cells in this compartment are proliferating, meaning that fluctuations in the niche compartment are entirely absent, with the only remaining fluctuations due to cells ejected from the niche compartment that subsequently divide in the differentiation compartment. Consistent with this explanation, we found that fluctuations in $N$ increased when more symmetric divisions in the niche compartment generated non-proliferating daughters (scenarios 1 and 2). Finally, we note that the well-established neutral drift model of symmetrically dividing stem cells in a niche of fixed size (*Lopez-Garcia et al., 2010*; *Snippert et al., 2010*) fails to reproduce the high symmetry in proliferation we experimentally observe between sister cells (*Figure 4—figure supplement 2E*), indicating that the size constraint of a niche is by itself not sufficient to generate this symmetry. In conclusion, our simulations show that the dominance of symmetric divisions we observed experimentally might function to minimize fluctuations in cell proliferation.

## Symmetry of proliferation is independent of Paneth cell distance

Our results raised the question how the strong symmetry in proliferative behavior between sister cells is generated. Stem cell maintenance and cell proliferation are controlled by signals such as Wnt and EGF, that in organoids are locally produced by Paneth cells (*Sato et al., 2011*; *Farin et al., 2016*). The symmetry between sister cells could therefore be explained by these sisters having a similar position relative to Paneth cells, leading them to experience a near identical environment in terms of proliferative signals. Alternatively, the proliferative behavior of sister cells could be controlled through

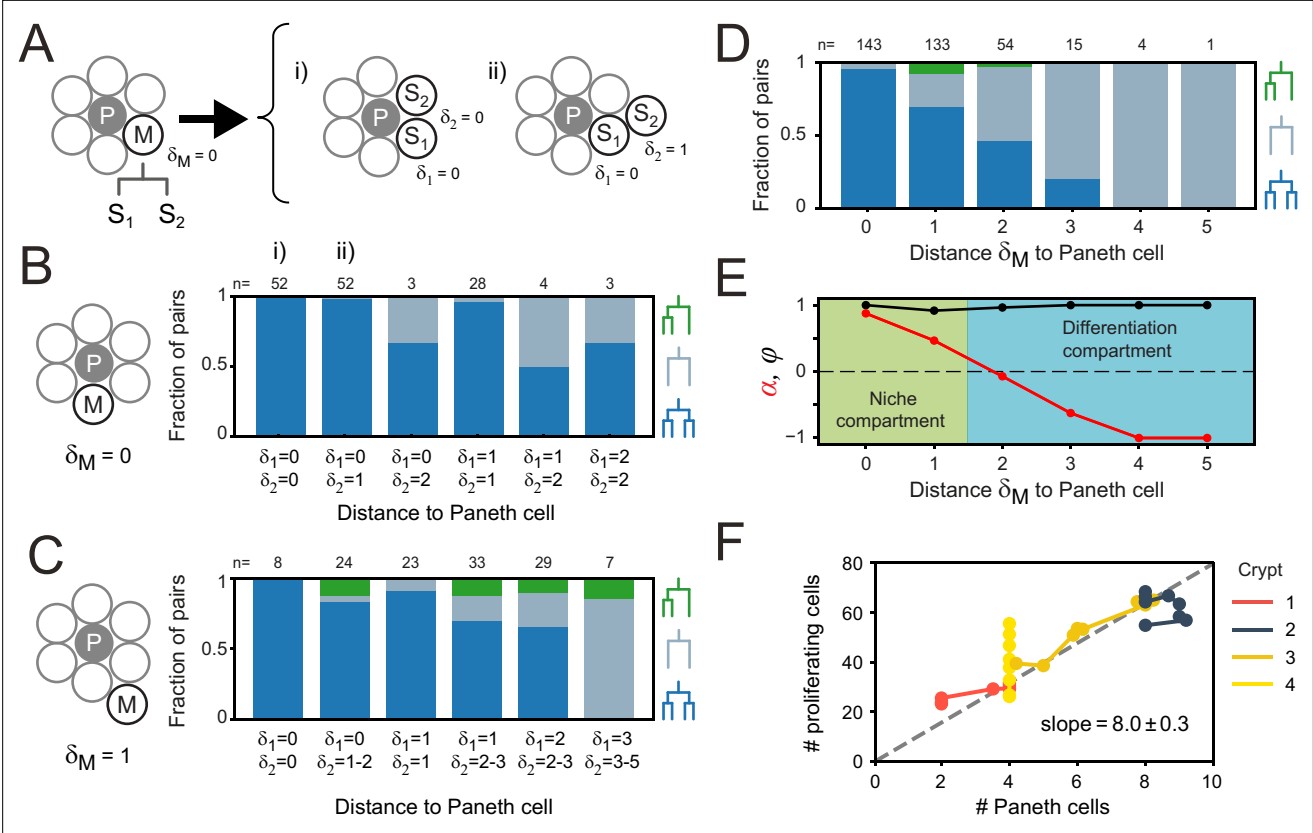

**Figure 5.** Impact of Paneth cell distance on proliferation. (**A**) Dependence of proliferation on contact with Paneth cells. We examined cases where a mother cell (**M**) that touched a Paneth cell (**P**) divided into sister cells, $S_1$ and $S_2$, that either retained or lost Paneth cell contact. Here, the link distances $\delta_1$ and $\delta_2$ represent the number of cells between each sister and its closest Paneth cell. (**B**) Probability that both cells divide (blue), neither cell divides (orange) nor only a single cell divides (green) for all sister pairs $S_1$ and $S_2$ of which the mother touched a Paneth cell ($\delta_M = 0$). Sister pairs exhibited full symmetry in proliferative behavior, even when distance to the Paneth cell differed between sisters ($\delta_1$). (**C**) Same as (**B**) but for a mother cell positioned one cell away from the Paneth cell ($\delta_M = 1$). More daughter cells cease proliferation. While the fraction of pairs where only one sister divides increases, most sisters exhibit symmetric behavior. (**D**) Probability of each division pattern as a function of Paneth cell distance of the mother cell. (**E**) Proliferative bias $\alpha$ and degree of symmetry $\Phi$ as a function of Paneth cell distance. The observed values of $\alpha$ define a proliferative niche (green, $\alpha \approx 1$) and non-proliferative differentiation (blue, $\alpha < 0$) compartment, with the former corresponding approximately to the first two 'rings' of cells surrounding the Paneth cell. (**F**) Number of proliferating cells as a function of Paneth cell number. Time courses for individual crypts were divided into 5-hr intervals (markers), for which average cell numbers were calculated. Apart from crypt 4, Paneth cell number correlated well with number of proliferating cells, even when Paneth cell number increased in time due to divisions. Dashed line is a linear fit to the data.

The online version of this article includes the following video and figure supplement(s) for figure 5:

**Figure supplement 1.** Proliferation dynamics as a function of Paneth cell distance.

**Figure 5—video 1.** Crypt unwrapping and Paneth cell distance.

https://elifesciences.org/articles/80682/figures#fig5video1

the lineage, by their mother. In this case, symmetric proliferative behavior would even be seen in sisters that differ in position relative to Paneth cells. To differentiate between these two scenarios, we performed lysozyme staining after time-lapse imaging to retrospectively identify Paneth cells in our tracking data. Using crypt 'unwrapping' (*Figure 1D and E*), we calculated for each cell and each time point the link distance δ to the closest Paneth cell (Materials and methods, *Figure 5—figure supplement 1A and B*, *Figure 5—video 1*), i.e., the number of cells between the cell of interest and its closest Paneth cell, allowing us to examine proliferative behavior as function of distance to Paneth cells.

Paneth cell-derived Wnt ligands form gradients that only penetrate 1–2 cells into the surrounding tissue (*Farin et al., 2016*), suggesting that the steepest gradient in proliferative signals is found in close proximity to the Paneth cell. We therefore selected all dividing mother cells directly adjacent to a Paneth cell ($\delta_M = 0$) and examined their daughters. These sister pairs varied in Paneth cell distance

($\delta_{1, 2} \approx$ 0–2, *Figure 5A*, *Figure 5—figure supplement 1C*), with differences between sisters ($\delta_1$) seen in 42% of pairs. We classified each sister pair as asymmetric in outcome, when only one sister continued proliferating, or symmetric (*Figure 5B*). In the latter case, we distinguished between symmetric pairs where both sisters divided and those were both stopped proliferating. We found that most daughters cells divided again, even though a small fraction ceased division even in close proximity to Paneth cells (*Figure 5B*, *Figure 5—figure supplement 1D*). However, whether cells divided or not was fully symmetric between sister pairs, even when one cell remained adjacent to a Paneth cell ($\delta_1 = 0$) while the other lost contact ($\delta_2 > 0$). This also held for the few pairs where Paneth cell distance differed most between sisters ($\delta_1 = 0$, $\delta_2 = 2$).

When we instead examined mother cells that just lost contact with a Paneth cell ($\delta_M = 1$), we found that their offspring stopped proliferating more frequently (*Figure 5C*). While here we did find a substantial fraction of sister pairs with asymmetric outcome, for most pairs the outcome was still symmetric (92% of pairs), even for pairs that differed considerably in relative distance to the Paneth cell. Sister pairs with asymmetric outcome occurred more frequently for pairs with different Paneth cell distances ($\delta_1 \neq \delta_2$). For these pairs, however, the non-proliferating cell was the sister closest to the Paneth cell about as often as it was the more distant (five and three pairs, respectively), indicating that position relative to the Paneth cell had little impact on each sister's proliferative behavior. Overall, these results show that the symmetry of proliferative behavior between sisters did not reflect an underlying symmetry in distance to Paneth cells, thus favoring a model where this symmetry is controlled by the mother cell.

## Paneth cells control proliferative bias

Even though the proliferative behavior of sisters was not explained by their relative Paneth cell distance, we found that the bias toward proliferating daughters was clearly reduced when the Paneth cell distance of the mother increased (*Figure 5B and C*). Our simulations showed that both division symmetry and proliferative bias are important parameters in controlling fluctuations in the number of proliferating cells, with fluctuations minimized when most divisions are symmetric ($\Phi \approx 1$), biased strongly toward producing two proliferating daughters in one compartment ($\alpha \approx 1$) and two non-proliferating daughters in the other ($\alpha \approx -1$). To compare our experiments against the model, we therefore measured the frequency of each division class as a function of the mother's Paneth cell distance, averaging over all positions of the daughter cells (*Figure 5D*). Overall, cells had a broad range of Paneth cell distances ($\delta = 0$–10). Close to Paneth cells ($\delta \leq 1$), most divisions generated two proliferating daughters, while further away ($\delta > 1$), the majority yielded two non-proliferating cells. Asymmetry was rare and only occurred for $\delta = 1$–2. No divisions were seen for $\delta > 5$. When we used these measured frequencies to calculate $\alpha$ and $\Phi$ as a function of Paneth cell distance (*Figure 5E*), we found good agreement with the parameter values that minimized fluctuations in the model, with a niche compartment of strong proliferation close to Paneth cells ($\alpha = 0.67$, $\delta \leq 1$) and a non-proliferative compartment beyond ($\alpha = -0.67$), while almost all divisions were symmetric ($\phi = 0.98$).

The compartment model also predicted that the number of proliferating cells increases linearly with size $S$ of the niche compartment. Above, we observed that the number of proliferating cells differed between crypts (*Figure 2B*). We therefore examined whether variation in number of proliferating cells between crypts could be explained by differences in Paneth cell number, in those crypts where we identified Paneth cells by lysozyme staining (crypts 1–4). For the crypts that maintained a constant number of proliferating cells in time (crypts 1–2), we found that differences in number of proliferating cells were well explained by differences in Paneth cell number. Moreover, in crypts with increasing number of proliferating cells (crypts 3–4), we found that for crypt 3, this change could be explained by an increase of Paneth cell number, due to cell divisions that generated Paneth cell sisters. In crypt 4, however, proliferating cells increased in number without apparent Paneth cell proliferation. This crypt appeared to undergo crypt fission (*Langlands et al., 2016*) at the end of the experiment, suggesting that during fission cell proliferation is altered without concomitant changes in Paneth cell number. For crypts 1–3, the relationship between number of proliferating and Paneth cells was well fitted by a linear function (*Figure 5F*), consistent with the compartment model. The fitted slope of this line indicates that one Paneth cell maintains approximately eight proliferating cells. This agrees with the observation in *Figure 5D* that divisions are strongly biased toward proliferation only for cells within the first and second ring of cells around each Paneth cell. Taken together, these results show

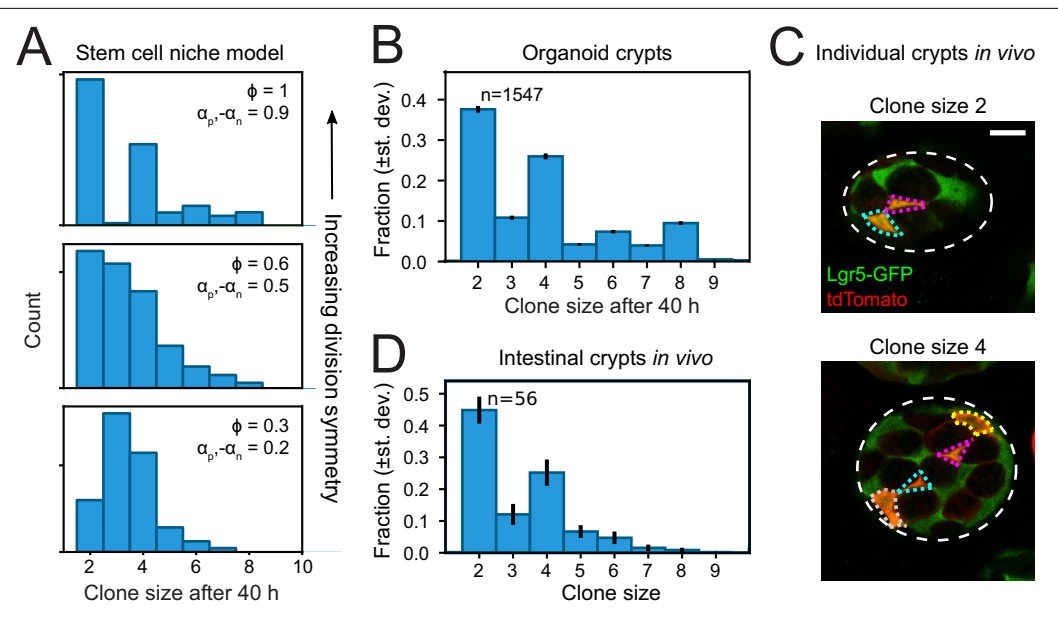

**Figure 6.** Clone size distributions reveal symmetry of proliferative behavior. (**A**) Clone size distributions calculated for the Compartment model, for different degrees of division symmetry $\Phi$. Top panel corresponds to parameters that minimize fluctuations in number of proliferating cells. For high division symmetry, $\Phi$, even clone sizes are enriched compared to odd clone sizes. (**B**) Clone size distributions calculated for the lineage data obtained in organoids, for a sliding window of 40 hr. Even clone sizes are enriched, consistent with the observed dominance of divisions with symmetric proliferative outcome. Error bars indicate standard deviation calculated using a bootstrapping approach (Materials and methods). (**C**) Examples of individual crypts found in vivo, displaying clone size 2 (top) and 4 (bottom). Crypts are viewed from the bottom, with individual cells belonging to a tdTomato+ clone (red) outlined. Scale bar is 10 μm. (**D**) Clone size distributions measured in vivo 60 hr after induction of Cre-mediated recombination in small intestinal crypts of $Lgr5^{EGFP\text{-}ires\text{-}CreERT2}$;$R26_{LSL\text{-}tdTomato}$ mice (n=160 crypts). Even clone sizes 2 and 4 are enriched compared to odd clone sizes 3 and 5. Error bars indicate standard deviations calculated using a bootstrapping approach.

that Paneth cells control proliferation by tuning the proliferative bias of divisions that are otherwise symmetric in proliferative outcome.

## In vivo lineage tracing confirms symmetric proliferative behavior of sister cells

Finally, we asked whether the symmetry of proliferative behavior between sister cells was also observed in intestinal (stem) cells in vivo. Studying lineage dynamics with the high spatial and temporal resolution we employed here is currently impossible in vivo. However, we found that clone size distributions, which can be measured in vivo, exhibited a clear signature consistent with symmetric divisions. Specifically, clone size distributions of lineages generated by the compartment model showed that enrichment of even-sized clones depended strongly on a high frequency of symmetric divisions (*Figure 6A*). When we quantified clone size distributions for our organoid lineage data, by counting the number of progeny of each cell at the end of a 40-hr time window, while sliding that window through our ~60-hr data set, we indeed found that even clone sizes were strongly enriched compared to odd clone sizes (*Figure 6B*). Both for organoid and model data, we still observed odd clone sizes even when virtually all divisions were symmetric in proliferative behavior. This reflected variability in the cell cycle duration, with odd clone sizes typically resulting from symmetric divisions where one daughter had divided, but the other not yet.

To measure clone size distributions in the small intestine in vivo, we stochastically induced heritable tdTomato expression in Lgr5 + stem cells using $Lgr5^{EGFP\text{-}ires\text{-}CreERT2}$;$R26_{LSL\text{-}tdTomato}$ mice. We activated Cre-mediated recombination by tamoxifen and examined tdTomato expression after 60 hr, similar to the timescale of our organoid experiments, and imaged crypts with 3D confocal microscopy. Cre-activation occurred in one cell per ~10 crypts, indicating that all labeled cells within a crypt comprised

a single clone. Indeed, we typically found a small number of tdTomato + cells per crypt, of which most also expressed Lgr5-GFP (*Figure 6C*). We then counted the number of tdTomato + cells per crypt to determine the clone size distribution and found a clear enrichment of even clone sizes (*Figure 6D*), with the overall shape of the distribution similar to that measured in organoids. Overall, these results indicated a dominant contribution of divisions with symmetric proliferative outcome also in the lineage dynamics of Lgr5 + stem cells in vivo.

## Discussion

Self-renewing tissues exhibit homeostasis at multiple levels, such as overall tissue morphology, total cell number, and the relative frequency of different cell types. To prevent exponential growth or tissue atrophy, the birth of each proliferating cell must be balanced by the loss of another through terminal differentiation. Experiments in a range of systems indicate that this is achieved through 'population asymmetry', with each cell making the decision to proliferate or not in a stochastic manner and this balance only achieved averaged over the entire population (*Simons and Clevers, 2011*; *Clayton et al., 2007*; *Snippert et al., 2010*; *Klein et al., 2010*). However, our simulations showed that even though 'population asymmetry' ensures a constant pool of proliferating cells on average, its inherently stochastic nature can cause strong fluctuations in proliferating cell number, even resulting in their full depletion (*Figure 4*). This raises the question how these fluctuations are controlled.

We addressed this by a combined experimental and theoretical approach. We tracked all cell movements and divisions in the crypts of growing intestinal organoids, to reconstruct the full lineage of these crypts up to six generations (*Figure 1*). These data showed that the number of proliferating cells in most organoid crypts was approximately stationary, with small, i.e., sub-Poissonian fluctuations in their number, while non-proliferating cells were born at a constant rate (*Figure 2*, *Figure 2—figure supplement 1*), an indication of homeostatic control of cell proliferation that also limits fluctuations. That intestinal organoids exhibited homeostasis is notable, as organoid culture completely lacks surrounding tissue, such as the mesenchyme, that provides key signals regulating stem cell fate and proliferation (*Farin et al., 2012*; *Shoshkes-Carmel et al., 2018*), and shows that this form of homeostasis is inherent to the epithelium itself.

Our simulations showed that the fluctuations in proliferating cell number depended strongly on the relative proportion of divisions with symmetric proliferative outcome (either two proliferating or two non-proliferating daughters) and asymmetric outcome (one proliferating and one non-proliferating daughter), with small, sub-Poissonian cell number fluctuations only seen when most divisions had symmetric outcome (*Figure 4*, *Figure 2—figure supplement 1*). So far, the relative contribution of these three divisions patterns in the intestine could only be inferred indirectly from static measurements, leading to conflicting results (*Lopez-Garcia et al., 2010*; *Snippert et al., 2010*; *Itzkovitz et al., 2012*; *Sei et al., 2019*). Here, we used direct measurements of cell dynamics in time to unambiguously identify the proliferative state of successive generations of cells. These measurements show that virtually all cell divisions (>90%) showed symmetric proliferative behavior, generating either two proliferating or two non-proliferating sisters (*Figure 3*). Clone size distributions calculated based on our measured lineage data in organoids showed that this symmetry in proliferative behavior between sister cells gave rise to an enrichment of even clone sizes (*Figure 6*). Using short-term lineage tracing experiments in the mouse small intestine, we found that single Lgr5 + stem cells also gave rise to more even-sized than odd-sized clones, indicating that divisions that are symmetric in proliferative behavior indeed also dominate stem cell proliferation in vivo.

The symmetry in proliferative behavior we observe between sister cells could arise because both cells experience a highly similar environment, in terms of proliferative signals, or rather indicate control of cell proliferation through the lineage, by the mother cell. The current models of stem cell dynamics in the intestinal crypt favor a strong role for position relative to the stem cell niche, formed in organoids by Paneth cells, and a minor role, if any, for control of cell proliferation through the lineage (*Lopez-Garcia et al., 2010*; *Snippert et al., 2010*; *Ritsma et al., 2014*). We found that sister cells exhibited symmetric proliferative behavior, even when sisters differed in distance to Paneth cells (*Figure 5*), the sole source of proliferative Wnt signals in intestinal organoids (*Sato et al., 2011*). This result implies control of proliferation by the mother cell rather than by each daughter's position in the stem cell niche. Our simulations provided a potential function for the predominance of divisions with symmetric proliferative outcome. When the tissue was subdivided into compartments of low and

high cell proliferation, with the latter resembling the stem cell niche, we found that fluctuations in the number of proliferating cells were virtually eliminated, provided that cell divisions were symmetric, with all divisions generating two proliferating daughters in the niche compartment and two non-proliferating daughters outside (*Figure 4*). Consistently, in our experiment we found that frequency of mother cells generating two proliferating rather than two non-proliferating daughters decreased with the mother's distance to the closest Paneth cell (*Figure 5*). Taken together, our results suggest a model where differences in proliferative behavior emerge in the cell lineage over at least two generations: while a mother cell division generates two daughters with the same proliferative behavior, these daughters might subsequently generate grand-daughters that differ in proliferative behavior, depending on each daughter's position relative to the Paneth cells. This is consistent with our observation that the symmetry of proliferative behavior between cousins is reduced significantly compared to sisters (*Figure 3*).

We used mathematical modeling to explore the dependence of fluctuations in cell proliferation on the degree of symmetry in cell division outcome, arriving at a two-compartment model that matched key features of our experiments (*Figure 4D–F*). It reproduced the observed low, sub-Poissonian fluctuations in number of proliferating cells (*Figure 2—figure supplement 1D*), but only when division symmetry was high, as we also observed experimentally. In contrast, high symmetry increased fluctuations in a spatially uniform stem cell model (*Figure 4B and C*), while a standard neutral drift model of a stem cell niche (*Lopez-Garcia et al., 2010*; *Snippert et al., 2010*) failed to reproduce the observed symmetry in outcome (*Figure 4—figure supplement 2*). Our model also reproduced the observed correlation in proliferative state between cousin cells (*Figure 4—figure supplement 2*), explaining it as arising from closely related cells having similar location in the tissue and therefore similar probability of leaving the stem cell niche. Finally, it predicted the observed division of the tissue in a compartment where most divisions generated proliferating cells (close to Paneth cells) and one where divisions mostly generated non-proliferating daughters (away from Paneth cells) (*Figure 5*). However, the simplified nature of our model also poses limits. First, the observed transition from proliferating to non-proliferating daughter cells was more gradual than predicted by the model, indicating that each divisions proliferative outcome depended on space in a more complex manner than captured by the model. Second, the existence of compartments and the degree of symmetry in division outcome are imposed externally by the model rules. It will be interesting to examine whether simple mathematical models can explain how these properties emerge from the internal cellular states, long-range signaling pathways, and local cell-cell interactions involved in intestinal homeostasis (*Simons and Clevers, 2011*; *Gehart and Clevers, 2019*).

Precise control of cell proliferation is key to homeostasis. It has been proposed that cells may sense cell density, either by chemical signals or mechanical cues, and decrease cell proliferation (known as contact inhibition) if the cell number is too high (*Lander et al., 2009*; *Sun and Komarova, 2012*; *Eisenhoffer and Rosenblatt, 2013*), thereby ensuring homeostasis of and minimize fluctuations in the number of proliferating cells. Here, we provide a mechanism that achieves this without explicit sensing of cell density. Instead, it relies on the dominance of divisions symmetric in proliferative behavior of the daughter cells, coupled with the organization of a tissue in a proliferative niche (stem and TA cell) compartment, and a non-proliferative differentiation compartment. Such an organization is found widely, e.g., in the skin, hair follicles, testis, among others (*Li and Xie, 2005*). In all these tissues, homeostasis of and minimizing fluctuations in the number of proliferating and differentiated cells must be essential. Hence, we speculate that the model we propose here, which exploits proliferative symmetry between sister cells to minimize fluctuations, is conserved more broadly and relevant to diverse tissue systems.

## Materials and methods

**Key resources table**

| Reagent type (species) or resource | Designation | Source or reference | Identifiers | Additional information |
|---|---|---|---|---|
| Biological sample (*Mus musculus*) | H2B-mCherry (intestinal organoids) | Other | - | Gift from Hubrecht Institute, Clevers group |

*Continued on next page*

*Continued*

| Reagent type (species) or resource | Designation | Source or reference | Identifiers | Additional information |
|---|---|---|---|---|
| Antibody | anti-lysozyme (polyclonal rabbit) | Dako | RRID:AB_2341230; Cat# A0099 | IF(1:800) |
| Antibody | anti-rabbit IgG H&L (Alexa Fluor 405) (polyclonal donkey) | Abcam | RRID:AB_2715515; Cat# ab175649 | IF(1:1000) |
| Chemical compound, drug | Advanced DMEM/F-12 medium | Life Technologies | Cat# 12634010 | - |
| Other | Wheat Germ Agglutinin (WGA), CF488 A Conjugate | Biotium | Cat# 29022 | 5 µg/ml |
| Other | RedDot1 Far-Red Nuclear Stain | Biotium | Cat# 40060 | 1:200 |

## Organoid culture

H2B-mCherry murine intestinal organoids were a gift from Norman Sachs and Joep Beumer (Hubrecht Institute, The Netherlands). Organoids were cultured in basement membrane extract (BME, Trevigen) and overlaid with growth medium consisting of murine recombinant epidermal growth factor (EGF 50 ng/ml, Life Technologies), murine recombinant Noggin (100 ng/ml, Peprotech), human recombinant R-spondin 1 (500 ng/ml, Peprotech), n-acetylcysteine (1 mM, Sigma-Aldrich), N2 supplement (1×, Life Technologies) and B27 supplement (1×, Life Technologies), Glutamax (2 mM, Life Technologies), HEPES (10 mM, Life Technologies), and Penicilin/Streptomycin (100 U/ml 100 µg/ml, Life Technologies) in Advanced DMEM/F-12 (Life Technologies). Organoid passaging was performed by mechanically dissociating crypts using a narrowed glass pipette.

## Time-lapse imaging

Mechanically dissociated organoids were seeded in imaging chambers 1 day before the start of the time-lapse experiments. Imaging was performed using a scanning confocal microscope (Nikon A1R MP) with a ×40 oil immersion objective (NA = 1.30). 30 z-slices with 2-µm step size were taken per organoid every 12 min. Experiments were performed at 37°C and 5% $CO_2$. Small but already formed crypts that were budding perpendicularly to the objective were selected for imaging. Imaging data was collected for three independent experiments.

## Fluorescent staining

After time-lapse imaging, organoids were fixed with 4% formaldehyde (Sigma) at room temperature for 30 min. Next, they were permeabilized with 0.2% Triton-X-100 (Sigma) for 1 hr at 4°C and blocked with 5% skim milk in Tris-Buffered Saline (TBS) at room temperature for 1 hr. Subsequently, organoids were incubated in blocking buffer containing primary antibody (rabbit anti-lysozyme 1:800, Dako #A0099) overnight at 4°C and then incubated with secondary antibody (anti-rabbit conjugated to Alexa Fluor405 1:1,000, Abcam #ab175649) at room temperature for 1 hr. Afterward, they were incubated with wheat germ agglutinin (WGA) conjugated to CF488 A (5 µg/ml Biotium) at room temperature for 2 hr, followed by incubation with RedDot1 Far-Red Nuclear Stain (1:200, Biotium) at room temperature for 20 min. Finally, organoids were overlaid with mounting medium (Electron Microscopy Sciences). The procedure was performed in the same imaging chambers used for time-lapse imaging in order to maintain organoids in the same position. Imaging was performed with the same microscope as previously described. Note that WGA stains both Paneth and Goblet cells, but the lysozyme staining allowed the unequivocal distinction between them.

## Single-cell tracking

Cells were manually tracked by following the center of mass of their nuclei in 3D space and time using custom-written image analysis software. Each cell was assigned a unique label at the start of the track. For every cell division, we noted the cell labels of the mother and two daughter cells, allowing us to reconstruct lineage trees. We started by tracking cells that were at the crypt bottom in the initial time point and progressively tracked cells positioned toward the villus region until we had covered all cells within the crypt that divided during the time-lapse recording. We then tracked at least one additional row of non-dividing cells positioned toward the villus region. Cell deaths were identified either by

the extrusion of whole nuclei into the organoid lumen or by the disintegration of nuclei within the epithelial sheet. Only crypts that grew approximately perpendicular to the imaging objective and that did not undergo crypt fission were tracked. During imaging, a fraction of cells could not be followed as they moved out of the microscope's field of view or moved so deep into the tissue that their fluorescence signal was no longer trackable. Because these cells were predominantly located in the villus region, where cells cease division, this likely resulted in the underestimation of non-proliferating cells. Data was discarded when a large fraction (>25%) of the cells in the crypt move out of the imaged volume.

## Classifying cell state

To classify cells as either proliferating or non-proliferating, we followed the following procedure. Defining proliferating cells was straightforward, as their division could be directly observed. As for non-proliferating cells, we applied two criteria. First, cells were assigned as non-proliferating when they were tracked for at least 30 hr without dividing. This was based on our observation that cell cycle times longer than 30 hr were highly unlikely (p=7.1·10$^{-7}$, from fit of skew normal distribution, *Figure 3—figure supplement 1*). However, we were not able to track all cells for at least 30 hr, as cells moved out of the field of view during the experiment or, more frequently, because they were born less than 30 hr before the end of the experiment. In this case, we defined a cell as non-proliferating if its last recorded position along the crypt axis was higher than 60 µm, as almost no divisions were observed beyond this distance (*Figure 2*). Finally, cells were assigned as dying based on their ejection from the epithelium, while the remaining unassigned cells were classified as undetermined and not included in the analysis.

We tested the accuracy of this approach as follows. In data sets of >60 hr in length, we selected the subset of all cells for which we could with certainty determine proliferative state, either because they divided or because they did not divide for at least 40 hr. We then truncated these data sets to the first 40 hr, which reduced the number of cells whose proliferative state we could identify with certainty, and instead determined each cells proliferative state based on the above two criteria. When we compared this result with the ground truth obtained from the >60-hr data sets, we found that out of 619 cells, we correctly assigned 141 cells as non-proliferative and 474 as proliferative. Only four cells were incorrectly assigned as non-dividing, whereas they were seen to divide in the >60-hr data sets.

## Estimation of significance of symmetric divisions

To estimate whether the experimentally observed fraction of sisters with symmetric division outcome could be explained by sister deciding independently to proliferate further or not, we used a bootstrapping approach. In our experimental data, we identified n=499 sister pairs, where the proliferative state of each cell was known and excluding pairs where one or both sisters died. In this subset of sisters, the probability of a cell dividing was found to be p=0.79. For N=10$^5$ iterations, we randomly drew n sister pairs, which each cell having probability p to be proliferative and *1* p to cease proliferation. For each iteration, we then calculated the resulting symmetry fraction $\Phi$. This resulted in a narrow distribution of $\Phi$ with average ± tandard deviation of 0.67±0.02, well separated from the experimentally observed value of $\Phi$=0.97. In particular, none of 10$^5$ iterations resulted in a value $\Phi \geq$ 0.97, leading to our estimated p-value of p<10$^{-5}$. Overall, this means that the high fraction of sisters with symmetric division outcome reflects correlations in sister cell fate.

## Estimation of crypt growth rate

To estimate an effective growth rate from the time dynamics of the total cell number and number of proliferating cells $N$ for each individual crypt, we used the 'Uniform' model as defined in the main text. Here, each generation the number of proliferating cells increases by $\alpha N$, and the number of non-proliferating cells, $M$, changes by $(1-\alpha)N$, where α is the growth rate with $-1 \leq \alpha \leq 1$. For α sufficiently close to zero, the resulting dynamics of the number of proliferating and non-proliferating cells, $N$ and $M$, is given by $\frac{dN}{dt}=\frac{\alpha}{T}N$ and $\frac{dM}{dt}=\frac{1-\alpha}{T}N$, where $T$ is the average cell cycle duration. Solving these differential equations yields $N(t) = N(0) exp\left(\frac{\alpha t}{T}\right)$ and $U(t) = M(0) - \frac{1-\alpha}{\alpha}D(0) + \frac{D(0)}{\alpha}exp\left(\frac{\alpha t}{T}\right)$ for the total number of cells, where $U=N+M$. We then fitted $U(t)$ and $N(t)$ to the experimental data in *Figure 2A and B*, using a single value of the fitting parameter $\alpha$ for each crypt and the experimentally determined value of T=16.2 hr.

## Crypt unwrapping

At every time point, the crypt axis was manually annotated in the $xy$ plane at the $z$ position corresponding to the center of the crypt, since tracked crypts grew perpendicularly to the objective. Three to six points were marked along the axis, through which a spline curve $s(r)$ was interpolated. Then, for each tracked cell $i$, we determined its position along the spline by finding the value of $r$ that minimized the distance $d$ between the cell position $x_i$ and the spline, i.e., $d(r_i) = min_r|s(r) - x_i|$. At each time point, the bottom-most cell of the crypt, i.e., that with the lowest value of $r_i$, was defined as position zero. Thus, the position along the axis $p_i$ for cell $i$ was defined as $p_i = r_i - min_i(r_i)$. To determine the angle around the axis $\theta_i$ for cell $i$, we considered a reference vector $u$ pointing in the direction of the imaging objective, given by $u = (0, 0, -1)$, and the vector $v_i = x_i - s(r_i)$ defined by the position of the cell $x_i$ and the position of minimum distance along the spline $s(r_i)$. Then, the angle is given by $\theta_i = acos(u \cdot v_i/uv)$.

## Distance to Paneth cells

To estimate the distance between cells we used the following approach. For each cell at each time point we found the five closest cells within a 15-µm radius, which became the edges in a graph representation of the crypt (*Figure 5—figure supplement 1*). These values were chosen because a visual inspection revealed an average nucleus size of 10 µm and an average of five neighbors per cell. This graph was then used to define the edge distance of a cell to the nearest Paneth cell. At every time point during the lifetime of that cell, the minimum number of edges required to reach the nearest Paneth cell was recorded. The edge distance is then defined as the number of edges minus one. For example, a neighbor cell of a Paneth cell (1 edge) has a distance of zero. When the edge distance of a cell to a Paneth cell varied in time, we used the mode of its distance distribution, i.e., the most frequently occuring value, as recorded during its lifetime.

## In vivo clonal tracing

All experiments were carried out in accordance with the guidelines of the animal welfare committee of the Netherlands Cancer Institute. $Lgr5^{EGFP-ires-CreERT2}$;$R26_{LSL-tdTomato}$ double heterozygous male and female mice (Bl6 background) were housed under standard laboratory conditions and received standard laboratory chow and water ad libitum prior to start of the experiment. 60 hr before sacrifice, mice received an intraperitoneal injection with 0.05 mg tamoxifen (Sigma, T5648; dissolved in oil) resulting in maximally 1 labeled cell per ~10 crypts. After sacrifice, the distal small intestine was isolated, cleaned, and flushed with ice cold phosphate-buffered saline (PBSO), pinned flat and fixed for 1.5 hr in 4% paraformaldehyde (PFA) (7.4 pH) at 4°C. The intestine was washed in Phosphate Buffered Solution/1% Tween-20 (1% PBT) for 10 min at 4°C after which it was cut into pieces of ~2 cm and transferred to a 12-well plate for staining. The pieces were permeabilized for 5 hr in 3% BSA and 0.8% Triton X-100 in PBSO and stained overnight at 4°C using anti-RFP (Rockland, 600-401-379) and anti-GFP (Abcam, ab6673) antibodies. After 3 times 30 min washes at 4°C in 0.1% Triton X-100 and 0.2% BSA in PBSO, the pieces were incubated with Alexa fluor Donkey anti rabbit 568 (Invitrogen, A10042) and Alexa fluor Donkey anti goat 488 (Invitrogen, A11055) secondary antibodies overnight at 4°C. After an overnight wash in PBT, the pieces were incubated with DAPI (Thermo Fisher Scientific, D1306) for 2 hr and subsequently washed in PBS for 1 hr at 4°C. Next, the intestinal pieces were cleared using 'fast light-microscopic analysis of antibody-stained whole organs' described in Messal et al. (*Sato et al., 2011*) In short, samples were moved to an embedding cassette and dehydrated in 30, 75, 2×100% MetOH for 30 min each at RT. Subsequently, samples were put into MetOH in a glass dish and immersed in methyl salicylate diluted in MetOH: 25, 75, 2×100% methyl salicylate (Sigma-Aldrich) 30 min each at RT protected from light. Samples were mounted in methyl salicylate in between two glass coverslips, and images were recorded using an inverted Leica TCS SP8 confocal microscope. All images were collected in 12 bit with ×25 water immersion objective (HC FLUOTAR L N.A. 0.95 W VISIR 0.17 FWD 2.4 mm). Image analysis was carried out independently by two persons. Afterward, all discrepancies between both datasets were inspected, resulting in a single dataset. Each biologically stained cell was annotated once in the 3D image. Different cells in the same crypt were marked as belonging to the same crypt, which is necessary to calculate the clone size for that crypt. Only crypts that were fully visible within the microscopy images were analyzed.

## Uncertainty estimation in clone size distributions

In organoids, the clone sizes are measured by calculating the number of offspring the cell will have 40 hr later. This calculation is performed for every hour of the time lapse, up to 40 hr before the end. In vivo, clone sizes are measured once per crypt, as we cannot view the dynamics over time. To estimate the uncertainty in our clone size distribution, both in organoids and in vivo, we use a bootstrapping approach. We denote the total number of clones observed as $N$. We then used random resampling with replacement, by drawing $N$ times a random clone from the data set of observed clones, to construct a new clone size distribution. We ran this procedure 100 times, each run storing the measured fraction of clones sizes. As a result, for every clone size we obtained a distribution of fractions, which we used to calculate the standard deviation of the fraction, as a measure of sampling error.

## Computational model

Simulations were initialized by generating a collection of proliferating cells, each belonging to either the niche or differentiation compartment. For each parameter combination, the initial number of proliferating cells assigned to each compartment was obtained by rounding to the closest integer the values given by the equations for $N_n$ and $N_d$ in the main text. When the initial number of proliferating cells in the niche compartment was lower than the compartment size $S$, they were randomly distributed over the compartment, with the remaining positions taken up by non-proliferating cells in order to fill the compartment. Each proliferating cell $c$ that was generated was assigned a current age $A_c$ and a cell cycle time $C_c$, i.e., the age at which the cell will eventually divide. The current age was obtained by randomly drawing a number from an interval ranging from 0 hr to the mean cell cycle time obtained from experimental data, while the cell cycle was obtained by drawing a random number from a skew normal distribution, which was fitted to the experimental distribution of cell cycle times as shown in *Figure 3—figure supplement 1*.

Simulations were performed by iterating the following routine over time, until a total simulation time $T = 10^6$ hr was reached. At each iteration $i$, we found the cell $c_i$ that was due to divide next, and a time step $\Delta t_i$ was defined by the time remaining for this cell to divide, i.e., $\Delta t_i = min_c \left( C_c - A_c \right)$. Then, the ages of all proliferating cells were updated, and the division of cell $c_i$ was executed. This was done by randomly choosing one of the three division modes defined in *Figure 4C*, according to the probabilities determined by the parameters $\alpha$ and $\phi$ of the compartment to which the cell belonged. Any proliferating daughters that were born were initialized with age zero and a random cell cycle time drawn. For the two-compartment model, if the proliferating cell belonged to the niche compartment, the distalmost cell within this compartment was transferred to the differentiation compartment, without changes to its proliferative state. This means that a proliferating cell that is transferred to the differentiation compartment will still divide, with the symmetry only determined by $\phi$, even if $\alpha_d = -1$, i.e., all divisions in the differentiation compartment generate non-proliferating daughters. This corresponds to the assumption that the decision to proliferate or not, as well as the symmetry between the resulting daughters, is set by the external environment (niche or differentiation compartments) the cell experiences at birth and cannot be reversed at a later point. Finally, the number of proliferating and non-proliferating cells in each compartment was updated accordingly. Cell rearrangements were implemented as follows. For each iteration $i$, with time step $\Delta t_i$, we drew the number of cell rearrangements from a Poisson distribution with mean $\left( r \cdot S \right) \Delta t_i$, where $r$ is the rearrangement rate per cell. We then implemented each individual rearrangement by randomly selecting a cell at position $j \in \left( 0, S\text{-}1 \right)$ and swapping it with the cell at position $j+1$.

The model had six parameters, of which three $(\alpha_n, \alpha_d, \phi)$ were systematically varied in our simulations. The remaining parameters were constrained by the experiments. We picked the niche size $S$ so that the total number of proliferating cells was 30, corresponding to the typical number of dividing cells observed in the experiments, through a procedure outlined in the main text. We obtained the average cell cycle duration $T$, as well as its distribution, from the data in *Figure 3—figure supplement 1*. Finally, we obtained the rearrangement rate $r$ from the observed (a)symmetry in proliferative fate observed between cousin cells. For a 'well-mixed' niche compartment, cousin pairs showed asymmetric outcome as often as symmetric outcome (*Figure 4—figure supplement 2*), in contrast to our experimental observations (*Figure 3A*). In contrast, for infrequent cell rearrangement, $r$, cells expeled from this compartment close together in time are also closely related by lineage, leading to

correlations in division outcome between cousins that reproduced those observed experimentally (*Figure 4—figure supplement 2*, *Figure 3A*).

For some parameter values, simulations were ended earlier than the total time $T$. This occurred when no proliferating cells were left in either compartment (defined as a depletion event), or when the number of proliferating cells reached an arbitrarily set maximum limit of five times its initial value (defined as an overgrowth event, occurring only in the one-compartment model). In these cases, simulations were restarted until a total simulation time $T$ was reached, and the total number of events was recorded. Thus, the rate of depletion or overgrowth refers to the number of times simulations had to be restarted for each value of $\phi$ divided by the total simulation time.

To obtain statistics regarding fluctuations on the number of proliferating cells $N$ through time, at each iteration $i$ we kept track of the number of proliferating cells in the niche compartment $d_i^p$ and in the differentiation compartment $d_i^n$. With these quantities, we could compute the standard deviation $\sigma$ of $N$ according to $\sigma^2 = \left\langle N^2 \right\rangle - \left\langle N \right\rangle^2$. Given that $N = N_n + N_d$, where $N_n$ and $N_d$ are the number of proliferating cells in the niche and differentiation compartments, $\sigma$ can be expressed as

$$\sigma^2 = \left\langle N_n^2 \right\rangle - \left\langle N_n \right\rangle^2 + \left\langle N_d^2 \right\rangle - \left\langle N_d \right\rangle^2 + 2\left\langle N_n N_d \right\rangle - 2\left\langle N_n \right\rangle \left\langle N_d \right\rangle, \text{ where } \left\langle N_{n,d} \right\rangle = \sum_i \frac{d_i^{n,d} \Delta t_i}{T} \text{ , } \left\langle N_{n,d}^2 \right\rangle = \sum_i \frac{\left(d_i^{n,d}\right)^2 \Delta t_i}{T}$$

and $\left\langle N_n N_d \right\rangle = \sum_i \frac{d_i^n d_i^d \Delta t_i}{T}$.

## Additional information

### Funding

| Funder | Grant reference number | Author |
| --- | --- | --- |
| Nederlandse Organisatie voor Wetenschappelijk Onderzoek | VIDI | Guizela Huelsz-Prince Yvonne Goos Jeroen S van Zon |
| Nederlandse Organisatie voor Wetenschappelijk Onderzoek | Building blocks of Life | Rutger Nico Ulbe Kok Xuan Zheng Sander Tans Jeroen S van Zon |
| Nederlandse Organisatie voor Wetenschappelijk Onderzoek | 680-47-529 | Guizela Huelsz-Prince Yvonne Goos Jeroen S van Zon |
| Nederlandse Organisatie voor Wetenschappelijk Onderzoek | 737.016.009 | Rutger Nico Ulbe Kok Xuan Zheng Sander Tans Jeroen S van Zon |

The funders had no role in study design, data collection and interpretation, or the decision to submit the work for publication.

### Author contributions

Guizela Huelsz-Prince, Data curation, Software, Formal analysis, Investigation, Visualization, Methodology, Writing - original draft; Rutger Nico Ulbe Kok, Data curation, Software, Formal analysis, Visualization, Writing - original draft; Yvonne Goos, Data curation, Investigation, Methodology; Lotte Bruens, Saskia Ellenbroek, Resources, Investigation, Methodology, Writing - original draft; Xuan Zheng, Data curation, Software, Visualization, Writing - review and editing; Jacco Van Rheenen, Sander Tans, Conceptualization, Supervision, Funding acquisition, Writing - review and editing; Jeroen S van Zon, Conceptualization, Formal analysis, Supervision, Funding acquisition, Writing - original draft, Writing - review and editing

### Author ORCIDs

Rutger Nico Ulbe Kok ⓘ http://orcid.org/0000-0002-6214-681X
Xuan Zheng ⓘ http://orcid.org/0000-0002-1700-9082
Jeroen S van Zon ⓘ http://orcid.org/0000-0002-6021-2924

### Ethics

All experiments were carried out in accordance with the guidelines of the animal welfare committee of the Netherlands Cancer Institute.

### Decision letter and Author response

Decision letter https://doi.org/10.7554/eLife.80682.sa1
Author response https://doi.org/10.7554/eLife.80682.sa2

---

## Additional files

### Supplementary files

• MDAR checklist

### Data availability

All cell lineage data, simulation code and data analysis scripts used to generate the figures have been deposited in Zenodo under accession codes https://doi.org/10.5281/zenodo.7197573.

The following dataset was generated:

| Author(s) | Year | Dataset title | Dataset URL | Database and Identifier |
|---|---|---|---|---|
| Huelsz-Prince G, Kok RNU, Goos YJ, Zheng X, van Zon JS | 2022 | Code and data from "Mother cells control daughter cell proliferation in intestinal organoids to minimize proliferation fluctuations" | https://zenodo.org/record/7197573#.Y24mQy2l1QJ | Zenodo, 10.5281/zenodo.7197573 |

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
