## [Editor Report]

This paper is a fundamental work in developmental biology that supports its findings with compelling evidence drawn from both theoretical and experiment insights. It provides a potentially general mechanism for the control of a proliferative cell population. This work will be of interest to researchers in the fields of developmental and stem cell biology.

---

## [Decision Letter]

**Decision letter after peer review:**

Thank you for submitting your article "Mother cells control daughter cell proliferation in intestinal organoids to minimize proliferation fluctuations" for consideration by *eLife*. Your article has been reviewed by 3 peer reviewers, one of whom is a member of our Board of Reviewing Editors, and the evaluation has been overseen by Didier Stainier as the Senior Editor. The following individual involved in the review of your submission has agreed to reveal their identity: Philip Greulich (Reviewer #3).

Essential revisions:

1) Please clarify the model definition such that there is no ambiguity regarding whether the compartments pertain to fate or proliferation (Reviewer 2).

2) Please clearly line out in the text how this model is to be interpreted by a readership that might not be familiar with mathematical modelling and in particular the way "simple" models as the one presented here relate to potentially more complex biology. Specifically, as Reviewers 1 and 3 point out, it needs to be discussed whether the compartment model should be interpreted at face value or whether it is a simplified mathematical description for potentially diverse biological scenarios.

*Reviewer #1 (Recommendations for the authors):*

The manuscript would benefit from a discussion of the scope of the findings, in particular the modelling results. The authors propose a compartmentalized model, which is to be understood as a coarse-grained mathematical description of a potentially diverse set of biological realities. The authors should be clearer in the discussion about what is the precise biological scope of the mathematical model.

L53: The final sentence of this paragraph is not clear. Presumably, the authors mean "fluctuations in *the number of* proliferating cells".

L97: The definition of proliferating cells of course strongly depends on the ratio of the typical cell cycle time and the time of duration of the experiment. Can the authors give a statistical estimate on the percentage of cells being misclassified as "non-proliferating" due to the finite length of the observation interval?

L104: It would help the readers if the authors briefly stated the basic assumptions underlying the "simple model of cell proliferation" in the main text.

L108: I was at first very confused about the naming of the variable D describing the number of proliferating cells. I would usually associate "D" with a diffusion coefficient with units m^2^/s while the number of cells is dimensionless. The authors might consider renaming this variable to "N".

Figure 2A, B: To my eyes, there are oscillations in the number of cells and partially the number of proliferating cells. Is this evidence for temporal synchronization of cell divisions?

Figure 2B: This figure would benefit from a statistical analysis of the slopes as it is not evident to me whether the majority of lines have a vanishing slope or positive slope.

Figure 2B, D: Could the enrichment of even clone sizes have arisen by chance (p-value)?

*Reviewer #2 (Recommendations for the authors):*

I have a few comments though on a few additional analyses/theoretical controls that might help the paper even more readable – especially as model-data comparisons are sometimes not the most straightforward in presentation, making me unsure I fully follow the reasoning in places.

Point A/ Figure 4: the text is in general well-written, although I had to re-read this a few times this part to make sure I fully understood the model. I feel like calling the two domains "proliferative compartments" vs "non-proliferative compartments" is a bit confusing, because in fact the compartments are not implemented through changing proliferation, but instead changing fate.

On the first read, I thought the model would be that the authors would simply turn on proliferation in the bottom compartment, and turn it off in the other. In fact, I wonder – given the fact that Figure 2 identifies each compartment by proliferation rate, whether it would be pedagogical for the reader to go through this model first (it can be in SI obviously). I guess the point of the author is that this model would fail vs data. But it would be interesting to see exactly how, to set the stage for their model (because as the best-fit requires anyways \α_p close to 1, this might give rise to similar predictions?). Plotting the model output in the same way as the previous data from Figure 2C-D might also be nice for comparison.

The last thing where I got confused by the model is that it implements compartments in a spatial way – if I understand correctly (line 637 of the SI). But the point of the authors, based on data, is that fate of daughters is dependent on lineage rather than position (although they say in lines 646-647 that the two are highly correlated in the model for the rT they choose – see also point below). It would be good to clarify this.

Point B: Also in Figure 4, it is not very clear how the authors pick their rearrangement rate rT (lines 188-191). They say that they pick rT=1 to reproduce well the correlation between cousins, but that they find similar findings for larger rT. Would this mean then that this parameter is not necessary for the model, and that this leads to over-fitting? (But then I got a bit confused as in the Supp part, they say large rT do no work? (Line 644-646). As the authors have done high-quality tracking, I would have thought that maybe they could "just" infer this parameter from the x,y,z, and t coordinates of neighboring cells. This would help distinguish model fitting and model prediction in a better way in the manuscript. In general, maybe the authors could use the modelling section on the SI to summarise a bit the fitting and prediction strategy (and maybe provide in Figure S5 a more systematic fitting/sensitivity analysis rather than showing two extreme values only of rT?).

Point C: Figure 6: Maybe the authors could define an "asymmetry index" (eg. the cumulative probability "missing" from the odd clones assuming some smooth gaussian from neutral drift without lineage correlations). I also wonder whether the authors could recapitulate their findings in a toy model of stochastic fate choices with correlation time T in outcome (to connect it more directly to previous models of neutral drift on a 1D ring). This type of model making correlation time in fates an explicit variable would go well with the discussion on the number of generations at which correlations in fate are lost (line 360-370). It would also be nice to mention the consequence that this would have on longer-term clonal conversion dynamics that have been extensively studied in the past.

*Reviewer #3 (Recommendations for the authors):*

As said in the public review, my concern is to take the two-compartment model too literally. I do agree that the presented analysis is fine, and the two-compartment model works as a valid simplification to capture the qualitative features of the mechanism, but I am worried that the reader takes this at face value. Instead, given the measurements in Figure 5E, it is more likely that the proliferative potential decreases continuously with distance from Paneth cells. So while the two-compartment model works as a simplification of the numerical analysis (being a representative of a larger class of models where proliferative potential decreases with distance), it does not work well as a description of reality.

Further suggestions, questions, and corrections:

– When saying that fractions of asymmetric divisions are "low" (e.g. line 135), then this should be compared with the case to be expected when sister cell fate is unrelated: namely in that case, asymmetric divisions would be at 50%.

– For α=0 one wouldn't expect uncontrolled exponential growth as stated in line 170. Stochastic fluctuations can be very large for α=0, even exceeding the set threshold of 5-fold, but this is still not exponential growth. It should also be mentioned that this may depend strongly on the arbitrarily set threshold.

– In line 195 it is said that "stochastic depletion occurred when α_p <~ 0.5". However, the scale in the referred figure is set arbitrarily. Since there the depletion rate is always non-zero, depletion will always occur after sufficiently large times. So when saying "stochastic depletion occurred when α_p <~ 0.5", then it should be said over which time scale of observation this is meant.

– In the caption of Figure 1 the part for panel G should refer to panel F instead.

– Figure 5: the colours are difficult to distinguish for a red-green colour impaired reader (roughly 10% of the male population): orange vs. green is difficult to distinguish and the thin black font colour vs. thin red font colour of vertical axis labels in Figure 5E are difficult to distinguish.

– The value of phi reported in Figure 5E (phi=0.98) is significantly higher than that reported in Figure 3A (adding the symmetric events when both sisters go on dividing, never dividing, or dying, gives phi = 0.81). Where does this discrepancy come from?

– Figure 6D: It would be helpful to have an interpolation curve to see the enrichment at n=6 (currently this is not visible). Alternatively, plotting on a logarithmic scale could make this more visible.

---

## [Author Response]

Essential revisions:1) Please clarify the model definition such that there is no ambiguity regarding whether the compartments pertain to fate or proliferation (Reviewer 2).

We now refer to ‘niche’ and ‘differentiation’ compartment, to emphasize that cells indeed change fate, from stem cell to differentiated cell, upon transitioning from one compartment to the next. For more details, see our response to Reviewer 2.

2) Please clearly line out in the text how this model is to be interpreted by a readership that might not be familiar with mathematical modelling and in particular the way "simple" models as the one presented here relate to potentially more complex biology. Specifically, as Reviewers 1 and 3 point out, it needs to be discussed whether the compartment model should be interpreted at face value or whether it is a simplified mathematical description for potentially diverse biological scenarios.

We addressed this now in the Discussion. For more details, see our response to Reviewer 3.

Reviewer #1 (Recommendations for the authors):The manuscript would benefit from a discussion of the scope of the findings, in particular the modelling results. The authors propose a compartmentalized model, which is to be understood as a coarse-grained mathematical description of a potentially diverse set of biological realities. The authors should be clearer in the discussion about what is the precise biological scope of the mathematical model.

We addressed this now in the Discussion. For more details, see our response to Reviewer 3, who raised similar issues.

L53: The final sentence of this paragraph is not clear. Presumably, the authors mean "fluctuations in the number of proliferating cells".

The Reviewer is correct, we changed this in the text.

L97: The definition of proliferating cells of course strongly depends on the ratio of the typical cell cycle time and the time of duration of the experiment. Can the authors give a statistical estimate on the percentage of cells being misclassified as "non-proliferating" due to the finite length of the observation interval?

Based on the measured distribution of cell cycle times, we can estimate that within 30 hours, 99.93% of all proliferating cells will have divided (Figure 3 —figure supplement 1B). For that reason, we classify cells for which we have not seen a division, and that we could track for at least 30 hours, as non-dividing. However, particularly towards the end of the experiment, cells cannot be followed for 30 hours, precluding assignment of non-proliferative fate based on that single criterium. We therefore employed a further criterium: any cell that is located at least 60 μm above the bottom of the crypt, is considered non-proliferating (as detailed in the Methods section ‘Classifying cell state’).

This leaves a fraction of cells that cannot be classified as (non)-proliferating (34%), either because they could not be tracked or because they were born to close to the end of the experiment to observe a division or exclude division based on the criteria above. The latter category is (by definition) particularly prominent towards the end of the experiment. By truncating our data set to cells born at least 15 hours before the end of the experiment we reduce the fraction of unclassified cells to 10%.

We had so far not explicitly tested the accuracy of our classification criteria. To do so we performed the following test: in >60 hour data sets, we selected all cells whose proliferative state we could determine with certainty, i.e. they either divided or did not divide after 40 hours, thereby representing a ground truth proliferation data set. We then truncated our >60 hour data sets to the first 40 hours, so that lineages of many of the cells in the ground truth data set were truncated as well. We then used the two above criteria (including position along the crypt-villus axis) to identify (non-)proliferative cells, and found that only 4/619 cells were misidentified compared to the ground truth data, being assigned as non-proliferative even though they were observed to divide in the ground truth data. Overall, this indicates that our procedure correctly assigns proliferative fate to the majority of cells.

Changes to the manuscript:

– We added a new panel (Figure 3 —figure supplement 1B) that shows the estimated probability of a proliferating cell having not yet divided after time T.

– We modified the text “To systematically study … cells to 10%” in the section “Control of cell proliferation in organoid crypts” in the Results to explain the criteria for classification of proliferation state and add information on the fraction of cells that cannot be classified based on these criteria.

– We edited the section ‘Classifying cell state’ in the Methods to more clearly explain the assignment procedure outlined above, and added the text “we tested the accuracy… data sets.” to describe the new analysis of the procedure’s accuracy.

L104: It would help the readers if the authors briefly stated the basic assumptions underlying the "simple model of cell proliferation" in the main text.

Changes to the manuscript:

– We edited the text “We then estimated … by α and 1-α per division, respectively” in the section “Control of cell proliferation in organoid crypts’ to clarify the underlying model assumptions.

– We edited the section ‘Estimation of crypt growth rate’ in the Methods to clarify the underlying model assumptions.

L108: I was at first very confused about the naming of the variable D describing the number of proliferating cells. I would usually associate "D" with a diffusion coefficient with units m^2^/s while the number of cells is dimensionless. The authors might consider renaming this variable to "N".

Originally, D stood for dividing cells, but we agree that in the current version this is not clear. We have therefore changed it to N, as suggested by the Reviewer. In the Methods section (“Estimation of crypt growth rate”), we previously used N for the total number of cells, which we now changed to U.

Figure 2A, B: To my eyes, there are oscillations in the number of cells and partially the number of proliferating cells. Is this evidence for temporal synchronization of cell divisions?

The observation is correct, but not seen in all organoids. We currently don’t think this synchronization is somehow regulated, but rather reflects the strong correlation we observe in the cell cycle times between mother and daughter cells (this data is not shown in the manuscript). Indeed, cells that divide at the same time are typically closely related. We speculate that in organoids where many cells divide at the same time, these might all have been generated from a single stem cell formed at the budding stage of the organoid, when only a small number of stem cells are present. Because of the correlation in cell cycle time, this stem cell’s daughters, granddaughters, etc. will still divide at similar times. Because the degree of synchrony varied strongly between organoids, we chose not to discuss it in the manuscript.

Figure 2B: This figure would benefit from a statistical analysis of the slopes as it is not evident to me whether the majority of lines have a vanishing slope or positive slope.

We have added error bars to the fitted values of α in Figure 2 —figure supplement 1C. These error bars indicate that for the majority of crypts α significantly deviates from zero, with most crypts showing a positive growth rate. We note that in the main text we do not claim that crypts show no growth (α=0), but rather that the growth rate is low, i.e. the change in number of dividing cells is small on the timescale of the average cell cycle duration.

Changes to the manuscript:

– We added error bars to Figure 2 —figure supplement 1C.

– We added the sentence “For most … was low” to the caption of Figure 2 —figure supplement 1, to emphasize that the growth rate deviates from zero for most crypts, even though the deviation is small.

– We added the sentence “Error bars are the standard deviation in the fit of α” to the caption of Figure 2 – supplementary figure 1.

Figure 2B, D: Could the enrichment of even clone sizes have arisen by chance (p-value)?

We used bootstrapping to calculate the standard deviations of the fraction of clones of size 2, 3, etc. The resulting error bars show that both for organoids (Figure 5B) and in vivo crypts (Figure 5D) the differences between even and odd-size clones are significant, apart from the difference between clone size 5 and 6 for in vivo crypts.

Changes to the manuscript:

– We changed Figure 5B,D, to display the histogram with fraction of clones rather than number of clones on the y-axis, and add error bars indicating standard deviation. We edited the caption to explain how the standard deviation was calculated.

– We added a section ‘Uncertainty estimation in clone size distributions’ to the Methods to explain the procedure we followed.

Reviewer #2 (Recommendations for the authors):I have a few comments though on a few additional analyses/theoretical controls that might help the paper even more readable – especially as model-data comparisons are sometimes not the most straightforward in presentation, making me unsure I fully follow the reasoning in places.Point A/ Figure 4: the text is in general well-written, although I had to re-read this a few times this part to make sure I fully understood the model. I feel like calling the two domains "proliferative compartments" vs "non-proliferative compartments" is a bit confusing, because in fact the compartments are not implemented through changing proliferation, but instead changing fate.

Our original reasoning for the compartment labels, is that for α_p>0 (‘proliferative compartment’) most divisions generate proliferating cells, while for α_n<0 (‘non-proliferative compartment’) most divisions generate non-proliferating cells. However, because of the nature of our simulation rules, for all values of -1<α<0 individual dividing cells can still be found in the non-proliferative compartment, which we agree is confusing.

We agree with the Reviewer that in our model cells do not always change from proliferating to non-proliferating upon changing compartment, but rather from one cell fate/type to another: from proliferating stem cells to non-proliferating differentiated cells. We therefore changed our wording to ‘stem cell niche’ or ‘niche’ compartment (α>0) and ‘differentiation compartment’ (α<0) throughout the paper.

Changes to the manuscript:

– We now used ‘(stem cell) niche’ or ‘differentiation’ compartment throughout the paper.

– We use the subscripts _n and _d to refer to these two compartments, e.g. α_n>0 and α_d<0.

On the first read, I thought the model would be that the authors would simply turn on proliferation in the bottom compartment, and turn it off in the other. In fact, I wonder – given the fact that Figure 2 identifies each compartment by proliferation rate, whether it would be pedagogical for the reader to go through this model first (it can be in SI obviously). I guess the point of the author is that this model would fail vs data. But it would be interesting to see exactly how, to set the stage for their model (because as the best-fit requires anyways \α_p close to 1, this might give rise to similar predictions?).

We have implemented the suggested model (which is equivalent to the ‘neutral drift on a 1D ring’ models mentioned by the Reviewer further below) as follows: we assumed a niche compartment of fixed size S. Every cell in the niche compartment will proliferate. However, after every division, a single randomly selected cell is removed from the niche compartment and thus halts proliferation. We find that this model fails to reproduce the experimental data in a number of ways (Figure 4 – supplementary figure 2E). Most prominently, asymmetric sister fate is as likely as symmetric fate (phi=0.5), independent of niche size (the only model parameter), in contrast to the high symmetry (phi=0.97, Figure 3A) observed experimentally.

It might be possible to extend this model, e.g. by introducing correlations between sister cells through spatial organization (as in Figure 4 – supplementary figure 2C,D) or by introducing a correlation time in fate transitions (as suggested by the Reviewer below), in such a way that it matches the experimental data more closely. However, we want to point out that the aim of our model was *not* to construct the simplest model that reproduces our experimental data (including the value of phi), but rather to construct the simplest realistic model in which we can vary phi explicitly, to systematically study the impact of changing division symmetry on the key properties of the model.

Changes made to the manuscript:

– We added the sentence “Finally, we note … generate this symmetry” to the section “Symmetry between sisters minimizes fluctuations in a cell proliferation model” in the Results, that mentions the mismatch between the neutral drift model and the experiments.

– We added the panel Figure 4 – supplementary figure 2E showing the results of the neutral drift model and added a caption that explains the model and interprets its results in more detail.

Plotting the model output in the same way as the previous data from Figure 2C-D might also be nice for comparison.

We use Figure 2C,D to show that in the experiments the proliferating region has fixed size in time (Figure 2C) and similar size between crypts (Figure 2D). However, in simulations the size of the proliferative/niche region is fixed, and has a very different (1D) geometry compared to the more complex (2/3D) geometry in organoids. Moreover, in our simulations we don’t take space into account for the non-proliferative/differentiation region (which is show in Figures 2C,D). Overall, this means we cannot plot simulation data as in Figure 2C,D and compare it to that figure in a meaningful way.

The last thing where I got confused by the model is that it implements compartments in a spatial way – if I understand correctly (line 637 of the SI). But the point of the authors, based on data, is that fate of daughters is dependent on lineage rather than position (although they say in lines 646-647 that the two are highly correlated in the model for the rT they choose – see also point below). It would be good to clarify this.

The experiments indeed show that sister cells typically share the same fate (Figure 3A), consistent with their fate being dependent more on lineage. In the model this is purely determined by the parameter phi. For phi=0.95 (high symmetry), the model reproduces this correlation in fate of sisters by definition, independent of the value of rT.

The reason for incorporating space within the proliferative/niche compartment (rather than having rT -> infinity, corresponding to selecting a random cell for transferal to the differentiation compartment) is the experimentally observed correlation between the fate of *cousins*, with most cousins showing the same fate (Figure 3A). If we randomly select a stem cell for transferal to the differentiation compartment (well approximated by rT=100), we find that the fate of cousins lacks correlation (Figure 4 – supplementary figure 2B,C), with most cousins showing different fate. This is because cousin fate is not hard-wired by an external parameter, like for sisters, but depends on the statistics by which each cell of a sister pair ends up being eject from the niche compartment. When cell mixing is decreased (e.g. rT=1), the probability of moving to the differentiation compartment depends strongly on the cell’s spatial position within the niche compartment, with sister cells typically being located at similar position. As a result, when one sister is ejected, typically the other sister is ejected soon after. In that case, their offspring will both have ‘differentiated’ fate. Indeed, for rT=1 the correlation between cousins is similar to what is seen experimentally (Figure 4 – supplementary figure 2B).

Changes to the manuscript:

– We edited the text “For the two-compartment … at a later point” in the section “Computational model” in the Methods, to clarify how lineage and space control the decision to proliferate or not.

Point B: Also in Figure 4, it is not very clear how the authors pick their rearrangement rate rT (lines 188-191). They say that they pick rT=1 to reproduce well the correlation between cousins, but that they find similar findings for larger rT. Would this mean then that this parameter is not necessary for the model, and that this leads to over-fitting? (But then I got a bit confused as in the Supp part, they say large rT do no work? (Line 644-646). As the authors have done high-quality tracking, I would have thought that maybe they could "just" infer this parameter from the x,y,z, and t coordinates of neighboring cells. This would help distinguish model fitting and model prediction in a better way in the manuscript.

This was indeed unclear. Only for sufficiently low rT can our simulation reproduce the observed correlation in fate between cousins (as outlined in the point above), see Figure 4 – supplementary figure 2B. This means that here our experimental observations properly constrain the rT parameter. However, the original sentence “…although we found similar results for higher *r*…” was incomplete. We meant to say that the magnitude of the fluctuations in cell number and their dependence on phi, α_n and α_d did not depend on the magnitude of r.

Because of the differences in geometry between simulation (1D) and experiments (2/3D) it is not straightforward to obtain a value for r directly from the experiments. However, rT=1 corresponds to one cell pair rearrangement per cell division. This is consistent with our observation that rearrangements tend to occur almost always when a (nearby) cell divides.

Changes to manuscript:

– We changed the sentence “although we found… strongly on r” in the section “Symmetry between sisters minimizes fluctuations in a cell proliferation model” in the Results, to explain that our conclusions on fluctuations do not depend on the value of rT.

In general, maybe the authors could use the modelling section on the SI to summarise a bit the fitting and prediction strategy (and maybe provide in Figure S5 a more systematic fitting/sensitivity analysis rather than showing two extreme values only of rT?).

As we explained above, we believe that the simulation data in Figure S5B (now Figure 4 – supplementary figure 2B) is sufficient for fitting rT=1.

We have added the text “The model had…those observed experimentally (Figure 4—figure supplement 2, Figure 3A)” to the section “Computational model” in the Methods, that summarizes together how we varied or constrained all model parameters.

Point C: Figure 6: Maybe the authors could define an "asymmetry index" (eg. the cumulative probability "missing" from the odd clones assuming some smooth gaussian from neutral drift without lineage correlations).

In principle, it is an interesting idea to compare an “asymmetry index” between experiments and models. However, as our model results in Figure 6A show, the exact value of such an index would depend in a complex manner on the exact model parameters. Comparing with a Gaussian function based on a neutral drift model is also challenging, in terms of interpretation. In our understanding, such Gaussians emerge in the limit of large clone size N as scaling functions (e.g. in Snippert, Cell 2010) of the form F(x), where x=N/<N>, which because of the dependence on <N> explicitly makes no statement on the relative contribution of even and odd clones.

Based on the comments of Reviewer #1, we have now added a statistical analysis of the measured clone size distributions, that shows that the enrichment of even clones is statistically significant.

I also wonder whether the authors could recapitulate their findings in a toy model of stochastic fate choices with correlation time T in outcome (to connect it more directly to previous models of neutral drift on a 1D ring). This type of model making correlation time in fates an explicit variable would go well with the discussion on the number of generations at which correlations in fate are lost (line 360-370). It would also be nice to mention the consequence that this would have on longer-term clonal conversion dynamics that have been extensively studied in the past.

It is an interesting question whether the experimentally observed correlations in fate/behavior of sister and cousin cells could be explained by an even simpler model than in Figure 4, by considering a ‘standard’ neutral drift model (i.e. a single compartment of fixed size) but with the decision to change fate (and thus leave the compartment) occurring with some correlation time. However, there appear to us to be many ways to implement such a type of model, with the details of implementation likely having major impact on the observed statistics. We therefore think that examining the behavior of such a different class of models would be an entire project in itself.

Reviewer #3 (Recommendations for the authors):As said in the public review, my concern is to take the two-compartment model too literally. I do agree that the presented analysis is fine, and the two-compartment model works as a valid simplification to capture the qualitative features of the mechanism, but I am worried that the reader takes this at face value. Instead, given the measurements in Figure 5E, it is more likely that the proliferative potential decreases continuously with distance from Paneth cells. So while the two-compartment model works as a simplification of the numerical analysis (being a representative of a larger class of models where proliferative potential decreases with distance), it does not work well as a description of reality.

We agree with the Reviewer that the function of the mathematical model is not described sufficiently clearly. The model’s main aim was to systematically examine the impact of changing the symmetry of fate (proliferative or not) between sister cells, as described by the parameter phi, on fluctuations in the number of proliferating cells. This was of course inspired by our experimental observation of strong symmetry between sisters, and allowed us to propose a potential function for this observed symmetry: minimization of fluctuations. Given the poor performance of the simplest, spatially uniform model (Figure 4B,C), we turned to the two-compartment model as the most basic extension of the uniform model.

Our aim was thus not to construct the ‘most realistic’ model and we did not explore models that would reproduce the experimentally observed symmetry without explicitly including a symmetry parameter like phi. Indeed, our experimental observations suggested a more continuous transition in proliferation, as the Reviewer points out. It is also possible that ‘simpler’ models, like the model with correlation time suggested by Reviewer #2, exist that can reproduce our experimental findings without explicitly including division symmetry as a parameter. We have now added a paragraph to the discussion that outlines the limits of our model, both in terms of fitting to the data and in explaining the origin of the division symmetry.

We want to point out that, in response to a point raised by Reviewer #2, we added simulation results of a simple ‘standard’ neutral drift model of symmetrically dividing stem cells with niche crowding control, i.e. for every cell division one cell is randomly selected to differentiate. This model fails to reproduce the observed symmetry between sisters cells (Figure 4 —figure supplement 2E). Hence, it is to us a priori not obvious that the class of models with negative feedback control, that Reviewer #3 refers to in the public review, can easily reproduce these results: it would be an interesting future direction for theoretical study.

Changes made to the manuscript:

– We added a paragraph “We used mathematical … intestinal homeostasis[1, 10]” to the Discussion, explaining key features of the experiments captured by the two-compartment model, as well as limits of our model.

– We added the sentence “We therefore examined … an external parameter” to the section “Symmetry between sisters minimizes fluctuations in a cell proliferation model” to explain more explicitly aim of the model.

– We added the sentence “Finally, we note … generate this symmetry” to the section “Symmetry between sisters minimizes fluctuations in a cell proliferation model” in the Results, to explain that the neutral drift models cannot explain the observed symmetry.

– We added the panel Figure 4 —figure supplement 2E showing the results of the neutral drift model and added a caption that explains the model and interprets its results in more detail.

Further suggestions, questions, and corrections:– When saying that fractions of asymmetric divisions are "low" (e.g. line 135), then this should be compared with the case to be expected when sister cell fate is unrelated: namely in that case, asymmetric divisions would be at 50%.

The Reviewer is correct that this is the number the fraction should be compared to. We have added the sentence “Indeed, if we … or not independently” to the section “Symmetry of proliferative behavior between sister cells” in the Results, to compare this fraction to what would be expected if the decision was uncorrelated between sisters.

– For α=0 one wouldn't expect uncontrolled exponential growth as stated in line 170. Stochastic fluctuations can be very large for α=0, even exceeding the set threshold of 5-fold, but this is still not exponential growth. It should also be mentioned that this may depend strongly on the arbitrarily set threshold.

We agree with both points the Reviewer raises here and made the following changes in response:

– We changed ‘uncontrolled exponential growth’ to ‘uncontrolled growth’.

– We added the sentence “Frequency of overgrowth depends strongly on the threshold value used” to the caption of Figure 4.

– In line 195 it is said that "stochastic depletion occurred when α_p <~ 0.5". However, the scale in the referred figure is set arbitrarily. Since there the depletion rate is always non-zero, depletion will always occur after sufficiently large times. So when saying "stochastic depletion occurred when α_p <~ 0.5", then it should be said over which time scale of observation this is meant.

We agree with the Reviewer that the depletion rate is always non-zero. We changed this sentence to “Stochastic depletion occurred at significant rate (>1 event per 10^3^ hours)” to make clear that this is an observation about the probability that this occurs.

– In the caption of Figure 1 the part for panel G should refer to panel F instead.

We corrected this.

– Figure 5: the colours are difficult to distinguish for a red-green colour impaired reader (roughly 10% of the male population): orange vs. green is difficult to distinguish and the thin black font colour vs. thin red font colour of vertical axis labels in Figure 5E are difficult to distinguish.

We changed the orange color to grey in Figure 5B-D and changed text color and size in Figure 5E.

– The value of phi reported in Figure 5E (phi=0.98) is significantly higher than that reported in Figure 3A (adding the symmetric events when both sisters go on dividing, never dividing, or dying, gives phi = 0.81). Where does this discrepancy come from?

We thank the reader for pointing out this discrepancy, which in the end took us more effort to fully resolve than we anticipated at first glance.

The first part of the explanation is that the number of phi = 0.81 quoted by the Reviewer is due to the inclusion of cell death. While we report the prevalence of cell death in Figure 3A, its occurrence seems highly complex. As can be seen in Figure 1F, it can be highly prevalent in specific, rapidly dividing lineages in the crypt, but not in other crypt lineages, suggesting the presence of heritable chromosomal defects. In addition, it is seen to occur, at much lower frequency, in non-proliferating cells in the villus, in a process that appears similar to cell extrusion as seen in vivo. Overall, this suggests to us that cell death is a process that is controlled rather independently from the decision to proliferate or not. For that reason, when calculating the experimentally observed value of phi, we ignore sister pairs in which one or two cells die. This led to a higher value for phi (0.92) than the one calculated by the Reviewer. This is also consistent with our simulations, that don’t include cell death, corresponding to the implicit assumption that if a cell wouldn’t have died, it would have likely adopted the same fate (proliferating or ceasing proliferation) as its sister.

However, this was still lower than the value of phi=0.98 reported for Figure 5E. Tracking down this difference eventually led us to uncover an error in calculating the fractions in Figure 3A. Specifically, we used a misplaced position of the crypt bottom. Combined with the rule that cells more than 60 μm from the crypt bottom are considered non-proliferative (see “Classifying cell state” in the Methods), this resulted in an erroneously high number of non-proliferative cells in the villus region. This error was only present in the analysis for Figure 3A and Figure 4 —figure supplement 2A and didn’t impact the other figures.

Correcting these numbers changed the fractions both for sisters and cousins in Figure 3A and Figure 4 —figure supplement 2A. Calculating phi based on the numbers in Figure 3A now gives phi=(0.59+0.15)/(0.59+0.15+0.02)=0.97, close to the number in Figure 5E. The remaining discrepancy is explained by the fact that in Figure 5E we use a smaller subset of organoids, in which Paneth cells were stained by lysozyme. That small difference between these different sets of organoids in itself indicates that this fraction is remarkably invariant between different organoids.

However, one consequence of correcting this error is that the fraction of non-proliferating sisters and cousins in Figure 3A and Figure 4 —figure supplement 2A has decreased substantially, from 0.4 to 0.15 for sisters. This also means that the reported ratio between symmetrically proliferating and non-proliferating sisters has decreased, from 0.38:0.4 to 0.59:0.15. At first sight, this now appears inconsistent with our observation in Figure 2A,B that the number of proliferating cells remains approximately constant in most crypts, as this would imply that this ratio should be close to 50:50.

However, we performed new analysis (Figure 3—figure supplement 2) that indicates that this apparent mismatch is likely an artefact of analysis choices made in constructing Figure 3A. As this figure focuses on the high symmetry of proliferative outcome between sisters, we included all sister pairs for which the proliferative state could be unambiguously classified, for all crypts. This includes crypts 3 and 4 that showed considerably growth in number of proliferating cells. Indeed, including only crypts with low growth rate already reduces this mismatch (Figure 3—figure supplement 2), even though it doesn’t fully resolve it. The second reason for this mismatch is that we excluded sister pairs where one or two cells were unclassified. As our experiments suggest that these unclassified cells are in majority non-dividing cells, Figure 3A thus likely undercounts non-proliferating sister pairs.

As we explain in more detail in the current version of the section ‘Control of cell proliferation in organoid crypts’, unclassified cells are either ‘untracked’, i.e. impossible to track due to imaging issues, or ‘undetermined’, meaning that they were born too close to the end of the experiment to be unambiguously identified as proliferating (division observed) or non-proliferating (no division observed for >30 h). Untracked cells, as can be seen for instance in Figure 1F, are typically found in the villus, in lineages that are unlikely to produce many proliferating cells. Undetermined cells are particularly prevalent in the final ~15 h of each time-lapse experiment, as many cells are born that will not be able to execute their cell division before the end of the imaging time window. However, if we constrain our analysis to cells that are born >15 h before the end of imaging (as we did in Figure 2A,B and now do in Figure 3—figure supplement 2), most cells that are undetermined are likely non-proliferating based on the measured cell cycle distribution (Figure 3—figure supplement 1). As a test, we calculated the ratio between proliferative and non-proliferative sisters under the most extreme assumption that all unclassified cells are non-proliferating. In this case, we indeed find a balanced ratio for proliferative and non-proliferative sisters in crypts with low growth rate (Figure 3—figure supplement 2B).

In response to our discovery of the analysis error in Figure 3A, we made two further changes:

First, in the previous version of the manuscript, we concluded, based on the incorrect fractions, that cells ceasing proliferation were more important for homeostasis than cell death. We still believe this is the case, but because of our likely underestimation of non-proliferating cells, as described above, we feel this conclusion is no longer supported by the data in Figure 3A, and we have therefore removed it.

Second, we examined whether the reduced fraction of non-diving cells in the new Figure 3A impact the statistical validity of our conclusion that sisters with symmetric outcome are overrepresented. We have now added analysis that the observed fraction of symmetric sisters cannot be explained by each sister making the independent decision to proliferate or not, and thus our conclusion that sister cell behavior is strongly correlated still firmly stands.

As an overall conclusion, the different values for the fractions in Figure 3A are still consistent with our observation of homeostasis in the number of proliferating cells in Figure 2, assuming that unclassified cells are predominantly non-proliferating as our experiments suggest. Moreover, the changes to Figure 3A do not impact in any way our paper’s main conclusion regarding the prevalence and potential function of high symmetry in division outcome between sisters.

Changes to the figures:

– We changed Figures 3A and 4 —figure supplement 2A to incorporate the correct fractions of sister and cousin pairs.

– We added the new Figure 3-supplement 2, that explains that the apparent mismatch between proliferating and non-proliferating sisters is likely due to the exclusion of sister pairs with unclassified cells, as these were probably in majority non-proliferating.

Changes to the Result section “Single-cell tracking of complete crypts in growing intestinal organoids”:

– We added the sentence “Finally, a small … scattering in the tissue” to point out more explicitly that we cannot track some cells, predominantly in the villus region.

Changes to the Results section “Control of cell proliferation in organoid crypts”:

– We edited the text “To systematically … cells to 10%” to explain the criteria for classification of proliferation state and add information on the fraction of cells that cannot be classified based on these criteria.

– We edited the sentence “Using this classification … for nine crypts”. In particular, we changed ‘number of cells’ to ‘number of cells born’, to emphasize that we are not tracking total cell number, but rather number of proliferating and non-proliferating cells. The birth of a cell that dies is therefore counted as an increase in number of non-proliferating cells rather than an decrease in total cell number. This is clarified also in the caption of Figure 2A.

Changes to Results section “Symmetry of proliferative behavior between sister cells”:

– We edited the sentence “To examine … S1 and S2” to clarify that Figure 3A focuses on correlations in proliferative behavior between sisters, not on the balance between proliferation and non-proliferation.

– We removed the sentence “We found that sister pairs showed cell death more rarely than division arrest in at least one of the sisters (16% vs. 54%), indicating that cell death played a minor role in balancing cell proliferation.”

– We changed the percentages to match the corrected values of the fractions in Figure 3A.

– We have added the sentence “Indeed, if we … was high (97%)” that discusses the 97% fraction when cell death is excluded.

– We added the sentence "and could not … bootstrap simulation, Materials and methods)" to describe the statistical test of the observed symmetry.

– We added the section “When examining all sister pairs … sisters (40%, Figure 3—figure supplement 2)” to describe the apparent mismatch between proliferating and non-proliferating sisters in Figure 3A and our explanation that this likely reflects that sisters with unclassified cells, which are excluded from Figure 3A, are predominantly non-proliferating sisters.

Changes to the Methods:

– We have added the sentence “During imaging … of non-proliferating cells.” to the section “Single cell tracking” to explain the likely origin of the discrepancy between proliferating and non-proliferating cells born.

– We added the section “Estimation of significance of symmetric divisions” to explain our bootstrapping approach.

– Figure 6D: It would be helpful to have an interpolation curve to see the enrichment at n=6 (currently this is not visible). Alternatively, plotting on a logarithmic scale could make this more visible.

We added a statistical analysis of the measured clone size distribution in Figure 6D (See response to Reviewer #1 for more details). This showed that the difference between n=5 and n=6 is not significant (i.e. error bars overlap), indicating that we have too few data points for n=6 to establish whether there is a significant enrichment compared to n=5 and n=7.

Other changes to the manuscript:

– We changed the references to SI figures to adhere to *eLife* conventions

– We made changes to adhere to *eLife* conventions for reporting statistical analysis.